# Terahertz waveform synthesis in integrated thin-film lithium niobate platform

Alexa Herter [1,4] ✉, Amirhassan Shams-Ansari [2,4] ✉,
Francesca Fabiana Settembrini[1], Hana K. Warner[2], Jérôme Faist[1],
Marko Lončar [2] & Ileana-Cristina Benea-Chelmus [3] ✉

Bridging the "terahertz gap" relies upon synthesizing arbitrary waveforms in the terahertz domain enabling applications that require both narrow band sources for sensing and few-cycle drives for classical and quantum objects. However, realization of custom-tailored waveforms needed for these applications is currently hindered due to limited flexibility for optical rectification of femtosecond pulses in bulk crystals. Here, we experimentally demonstrate that thin-film lithium niobate circuits provide a versatile solution for such waveform synthesis by combining the merits of complex integrated architectures, low-loss distribution of pump pulses on-chip, and an efficient optical rectification. Our distributed pulse phase-matching scheme grants shaping the temporal, spectral, phase, amplitude, and farfield characteristics of the emitted terahertz field through designer on-chip components. This strictly circumvents prior limitations caused by the phase-delay mismatch in conventional systems and relaxes the requirement for cumbersome spectral pre-engineering of the pumping light. We propose a toolbox of basic blocks that produce broadband emission up to 680 GHz and far-field amplitudes of a few V m$^{-1}$ with adaptable phase and coherence properties by using near-infrared pump pulse energies below 100 pJ.

The terahertz (THz) region (typically defined between 0.1 and 10 THz) has proven to be critical for numerous applications in both fundamental science and industry, including communications[1–3], sensing[4], non-invasive imaging[5], nanoscopy[4,6,7] and ultrafast classical and quantum systems. In particular, detecting THz radiation is central to spectroscopy since the THz range hosts low-energy material resonances. For example, THz radiation plays an ever-increasing role in security and diagnostic medicine, due to its non-invasive, non-ionizing[8] ability to distinguish between different types of materials with higher resolution than microwaves: metals, water, and various organic compounds. On the other hand, THz waves can be utilized to control elementary excitations such as electrons[9], spins[10], molecular motion[11] and photons[12] on shorter timescales than microwaves in miniaturized transducers that link THz systems with other systems. This speed advantage is crucial for the exploration of novel materials and their ultimate limits, such as low-dimensional, magnetic[13], Pockels[14] or superconducting[15] materials, e.g., before decoherence times. Finally, in communications, THz wireless links may leverage the possibility to employ narrow-angle, directional beams[16] as an information channel instead of wide-angle broadcasting that is nowadays the standard in microwaves.

The realization of these applications is contingent on the ability to control the temporal waveform of THz fields on sub-THz-cycle scales. Thus, providing means for their synthesis at will is of high importance. As a result, the next generation of THz systems requires custom-tailoring various features of the electromagnetic waves, including their

[1]ETH Zurich, Institute of Quantum Electronics, Zurich, Switzerland. [2]Harvard John A. Paulson School of Engineering and Applied Sciences, Harvard University, Cambridge, MA, USA. [3]EPF Lausanne, Hybrid Photonics Laboratory, Lausanne, Switzerland. [4]These authors contributed equally: Alexa Herter, Amirhassan Shams-Ansari. ✉e-mail: aherter@phys.ethz.ch; ashamsansari@seas.harvard.edu; cristina.benea@epfl.ch

amplitude, frequency and phase down to a single cycle of oscillation. Currently, the THz range is generally challenging to access both using optical and electronic technologies. Notable progress in the miniaturization of THz devices has been made in the area of quantum cascade lasers that now operate without the need for cryogenic gases[17,18]; however, their emission is in the upper range of the THz domain (1.5–100 THz). Furthermore, pulsed emission is generally tricky directly from a THz quantum cascade laser, but active mode-locking has been demonstrated[19]. High-frequency spintronic emitters[20] or optical Kerr combs[21,22] have generated stable THz radiation but with limited control over the frequency of the output wave.

A much more widely adopted method for the generation and detection of THz is to utilize second-order ($\chi^{(2)}$) nonlinear processes driven by compact and high-power ultrafast lasers in the near-infrared region. Optical rectification is a generation mechanism for THz radiation that exhibits many useful features and in particular has been extensively used for the generation of high electric field strengths. In this $\chi^{(2)}$ process, an intense laser beam generates a nonlinear polarization $P^{(2)}(t)$ following its intensity envelope $I_0(t)$. If the incident optical beam is a femtosecond pulse, the generated field can be a broadband THz pulse that contains few cycles[23,24]. Lithium niobate (LN)[24–27] has always been an excellent choice of material for THz generation, owing to its high nonlinear coefficient of $d_{33} = 27$ pmV$^{-1}$ [28], and low optical losses in the near-infrared. LN's high optical power handling is crucial for optical rectification since the power of the generated THz signal depends quadratically on the optical pump power.

To date, the THz generation schemes through optical rectification have been largely limited to bulk systems. In this area, several works stand out that achieve custom-tailoring of terahertz radiation, by using quasi-phase-matched crystals[29–33] and arbitrarily poled crystals[34] to tailor the frequency and amplitude of terahertz radiation, or polarization-shaped pump pulses to tailor the polarization of the emitted terahertz radiation[35].

The above bulk generation schemes are well suited for high-field generation but have several shortcomings. First, further custom-tailoring of the emission is achieved either through pre-conditioning of the pump pulse preceding the nonlinear crystal (e.g., through complex pump pulse shaping[36,37], external switches[38] or multi-pulse setups[39]) or post-conditioning of a broadband terahertz pulse after its emission into free space (by diffractive surfaces[40] or switches[41]). Second, the phase-matching properties inside the bulk crystals are fixed by the refractive index mismatch at THz and pumping frequencies in bulk crystals and cannot be controlled. This fact results in the emission of Cherenkov radiation at a fixed angle in bulk LN. Third, bulk LN crystals need to be pumped close to their surface to avoid the high absorption of THz radiation in this material, but ensuring minimal distance to the surface is challenging in bulk crystals that provide no fine-tuned control over the exact propagation of the pumping field. Finally, achieving sub-THz-cycle precision of various degrees of freedom in the waveform design is challenging as their high number would inevitably increase the complexity of bulk systems. Early work recognized that miniaturization may provide solutions to these issues, such as, e.g., ion slicing bulk crystals into thinner slabs[42]. With advances in nanostructuring TFLN[23], few theoretical proposals to generate THz radiation appeared recently[43], along with experimental proposals of using topological confinement in laser-written LN slabs[44].

In this work, we depart from the generation in bulk crystals and instead demonstrate chip-based generation of terahertz transients inside low-loss integrated photonic circuits in lithium niobate. We show that this implementation circumvents shortcomings of bulk systems and additionally provides uniquely versatile control over the temporal, spectral, phase, amplitude, and farfield characteristics of the generated waveform by design of chip-scale components alone (a detailed comparison of our work with various former approaches to generate arbitrary THz waveforms is provided in

Supplementary Note 6). We exploit several unique features of TFLN platform that cannot at all, or only hardly, be realized in bulk crystals. First, we employ rib waveguides that allow THz generation within a few hundred of nanometers close to the surface, minimizing its absorption. Second, our low-loss fabrication capabilities allow us to connect several basic components of this platform such as grating couplers, y-splitters, and combiners for the proposed arbitrary waveform synthesis on one single chip. Paired with the wide transmission window of LN and its high power resilience, this allows us to guide pump pulses of energy of 100 pJ through complex device architectures without significant losses or denaturation of the nonlinear material. Third, the monolithic patterning of THz antennas around the waveguides provides the missing ability to not only control the spectral and temporal properties of the emitted THz radiation, but also its emission pattern into the farfield by the design of the antenna alone. Finally, the ability to choose the waveguide dimensions allows for precise control over the effective mode index, the group index, as well as the field confinement of both the pump pulses and the terahertz field as they interact in the antenna region.

## Results
### Design concept

In our THz emission scheme, single mode waveguides on $x$-cut TFLN are used to guide sub-picosecond laser pulses of intensity $I_0(t)$ at a central wavelength of 1560 nm to the gap of THz gold bow-tie antennas as depicted in Fig. 1 and in more detail in insets I and II. The ultrashort pump pulses are polarized along the [001]-axis of LN lying the in the thin-film plane. Due to the high nonlinear tensor element $d_{33}$ in LN the propagating pulses induce a polarization $P^{(2)}(t)$ (inset II) that generates a local electric field amplitude d$E_{local}$ at all positions along the antenna gap:

$$dE_{local}(\Omega,y) = C_{gap} \cdot I_0(\Omega) \cdot \Omega \cdot \exp\left(-i\frac{n_g\Omega}{c}y\right)dy, \qquad (1)$$

which sums up to an overall field $E_{OR}(\Omega) = \int_0^{l_{gap}} dE_{local}(\Omega,y)$ acting as a source of THz radiation (Fig. 1, inset III, detailed deviation in Supplementary Note 2A). Here $C_{gap}$ is a constant that depends on the material properties, $\Omega = 2\pi f_{THz}$ the generated angular THz frequency, $I_0(\Omega)$ the Fourier transformation of the optical intensity envelope $I_0(t)$ and d$y$ is the length element along the pump propagation direction (derivation in Supplementary Note 2B). At low frequencies, the THz field profile is independent of the pulse length, therefore its amplitude increases linearly with frequency (compare Supplementary Fig. 4). In the antenna's gap, the portion of the THz field matching the resonance, described by a responsivity function $R(\Omega)$, is outcoupled from the waveguide into free space with a dipolar emission pattern (Fig. 1, inset III). The total outcoupled field is given by:

$$E_{THz}(\Omega) = E_{OR}(\Omega) \cdot R(\Omega). \qquad (2)$$

The resulting temporal waveform depends on the center frequency and linewidth of the antenna as well as the spectral and temporal characteristics of the pumping pulse. In TFLN platform, we can safely ignore THz absorption in LN due to the small propagation length of the emitted wave inside the 600 nm thick waveguide (see inset V and supporting transmission characterization of the bulk substrate in Supplementary Note 1C). Furthermore, an antenna gap of width $w_g = 3$ μm supports efficient generation along the gap while maintaining low absorption losses of the pump pulse caused by antenna electrodes (see Supplementary Note 1A, B). The phase delay associated with each position along the antenna gap where the terahertz is generated is linked to the group velocity of the optical pulse $v_g = \frac{c}{n_g}$ (simulation providing $n_g$ in Supplementary Note 1B).

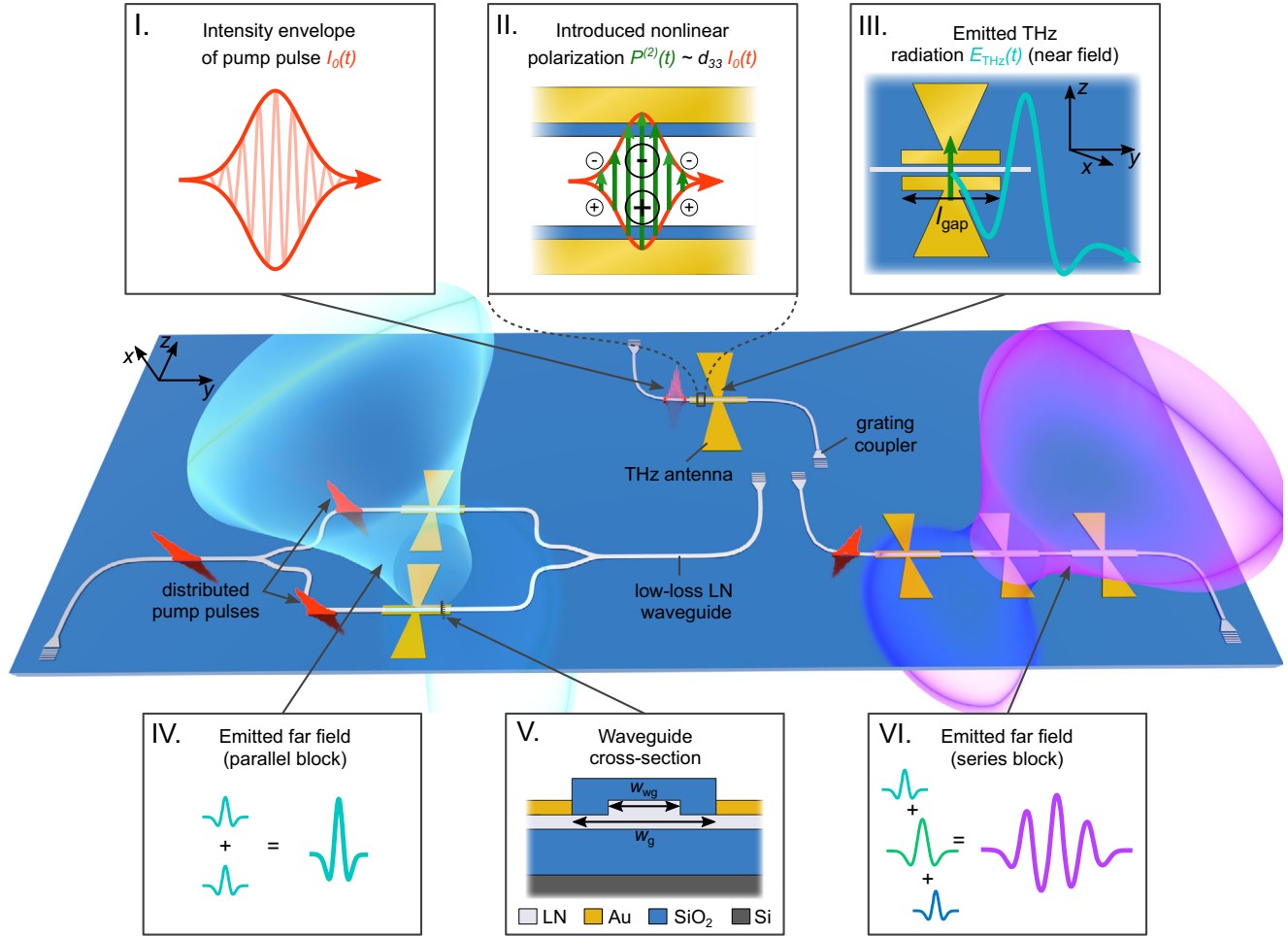

**Fig. 1 | Chip-based terahertz (THz) waveform synthesis from thin-film lithium niobate (TFLN) circuits.** A single THz emitter consists of an etched TFLN waveguide and a THz antenna (upper center chip design). Femtosecond pulses (shown in red) are coupled into the low-loss TFLN waveguides of width $w_{wg} = 1.5\,\mu m$ and pass through the two electrodes of a THz bow-tie antenna (shown in yellow) separated by a width $w_g$ (defined in inset V). THz radiation (shown in blue/purple) is generated through optical rectification in the antenna gap region delimited by parallel gold bars of length $l_{gap}$. The femtosecond pump pulse (inset I) generates a second-order polarization $P^{(2)}(t)$ that follows its envelope inside the $\chi^{(2)}$ waveguide (inset II). The nonlinear polarization inside the gap drives the emission of a broadband THz field $E_{THz}(t)$ into free space at the resonance frequency of the antenna (inset III). Several antennas may be combined to a single device. The layout

of the waveguides on chip and the relative position of antennas set the timing and the interference pattern in the farfield of THz emission. Parallel (lower left chip design) and serial (lower right chip design) blocks of antennas enable tailoring the temporal and spectral properties of the THz radiation. In the parallel configuration, the emitted waves from each antenna can interfere constructively and/or destructively following the phase-relation set by the relative distance between the two antennas in the farfield (inset IV). In the series configuration, the same pulse arrives at each antenna at different times. Each of the antennas generates a THz field of arbitrary shape and amplitude (denoted by colorful few-cycle THz pulses), resulting in various features in the THz emission field (inset VI). A cross-section of the generation region is provided (inset V, LN = lithium niobate, Au = gold, SiO₂ = silicon dioxide, Si = silicon).

In conclusion, the effective generation length can be written as:

$$l_{eff}(\Omega) = \left| \int_0^{l_{gap}} \exp\left(-i\frac{n_g\Omega}{c}y\right)dy \right| = l_{gap}\cdot\left|\,\mathrm{sinc}\left(\frac{n_g\Omega}{2c}l_{gap}\right)\right|. \quad (3)$$

$l_{gap}$ corresponds in the present study to a fraction of the THz field period (hence $l_{gap} < \frac{2\pi c}{n_g\Omega}$) for the antenna designs investigated (detailed derivation in Supplementary Note 2B). We define the coherence length as $l_{coh} = \frac{c}{4n_g f_{THz}}$.

## Distributed pulse phase matching

The low near-infrared losses of the TFLN platform[45] allow the pumping of several antennas using a single pulse. This provides an additional knob to tailor the THz emission, besides the individual antenna design parameters. We introduce a parallel and a serial architecture where one pulse is used to pump multiple devices (Fig. 1, bottom devices). In this scheme, the resulting THz waveform

depends strictly on the arrangement of antennas on-chip, their geometric distances and the group delay of the pumping pulses. This dependency can be summarized as a phase matching mechanism—to which we refer as distributed pulse phase matching—that is conceptually related to phased arrays with few-cycle pulses as the interfering fields (see Supplementary Note 2C). Distributed pulse phase matching is critical for the synthesis of arbitrary electric fields (the theoretical formalism is given in Supplementary Note 2D). In the parallel block, this enables constructive or destructive interference of THz pulses (inset IV), if antennas with identical parameters are employed. In the serial case, once the group delay $\tau_g$ of the pumping pulse between two serial antennas matches their resonant period $T_{THz}$ (and hence $\tau_g = \frac{1}{f_{THz}} = T_{THz}$), the generated pulses interfere constructively out-of-plane and increase the temporal coherence of the generated radiation (inset VI). Furthermore, the in-plane arrangement of antennas impacts the three-dimensional pattern of the emission, e.g., by increasing the aperture of generated THz light.

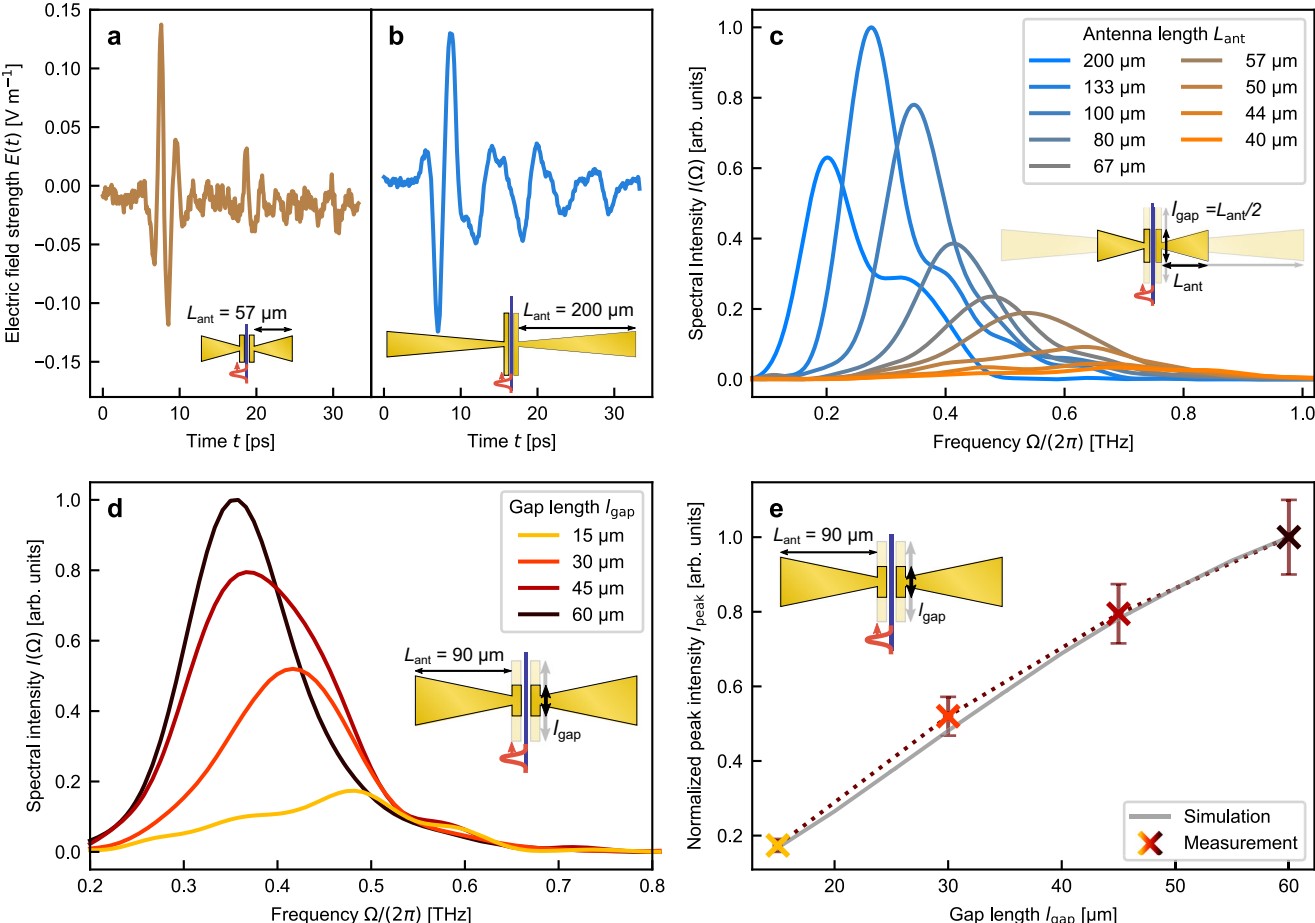

**Fig. 2 | Modifying temporal and spectral properties of the THz waveforms by antenna design. a**, **b** Time-domain trace of the electric field emitted by a single antenna device with an arm length $L_{ant} = 57\,\mu m$ and $L_{ant} = 200\,\mu m$, measured under identical experimental conditions. The antenna dimension, and its resonant frequency influence the characteristics of the emitted THz field: shorter antenna emits a few cycles of fast oscillations, whereas a longer antenna emits few slow oscillations. **c** Spectral intensities emitted by antennas of sizes $L_{ant} = 200\,\mu m$ to $L_{ant} = 40\,\mu m$ are retrieved from the Fourier transforms of the time-traces signals shown in **a** and **b**. The line color is changing from orange to blue for increasing $L_{ant}$.

Varying the size of a single antenna allows us to shift the emission between 180 and 680 GHz. **d** The antenna gap length ($l_{gap}$) influences the amplitude of THz emission as it delimits the generation region, as shown by the normalized spectral intensity emitted by antennas of $l_{gap} = 15$–$60\,\mu m$ and $L_{ant} = 90\,\mu m$. The line color is changing from yellow over red to black for increasing $l_{gap}$. **e** Measured values of the peak spectral intensities in (**d**) (crosses, linearly interpolated with dotted line) and simulated values (gray solid line) as a function of the gap length $l_{gap}$. The error bars of the measured peak intensity are estimated from the observed signal changes caused by limited alignment precision. $\Omega$ = THz angular frequency.

## Experimental investigation

Here, we experimentally tailor the THz radiation by designing chip-scale structures and generating THz waveforms with different properties enabled by distributed pulse phase matching. We employ a dual-wavelength THz time-domain setup to measure the electric field emitted by our devices on sub-THz-cycle timescales (description of setup in the Methods section "Optical setup" and Supplementary Note 5A).

First, we experimentally tailor the temporal and spectral properties as well as the amplitude of THz waveforms only by sweeping the antenna design parameters (Fig. 2, see also Supplementary Table 1). In particular, we study the impact of the antenna length $L_{ant}$, and gap length $l_{gap}$. $l_{gap}$ delimits the generation length by the geometrical dimensions of the two parallel gold bars. $L_{ant}$ determines the antenna resonance frequency and the temporal shape of the emitted THz fields. The measured time-trace of the THz electric field displays several rapid cycles for the shorter antenna ($L_{ant} = 57\,\mu m$) compared to the larger antenna ($L_{ant} = 200\,\mu m$) that rather emits a strong pulse with a slower cycle (Fig. 2a, b). The latter is followed by low-frequency oscillations that are characteristic of the antenna's lower resonant frequency. The peak-to-peak electric field measures in both cases roughly $0.25\,Vm^{-1}$. In the frequency domain, the initial portion of the THz pulse in Fig. 2b

corresponds to a broad resonant background spanning up to 350 GHz that is typical for bow-tie antennas, whereas the lasting oscillations afterward can be associated with the resonance frequency of the antenna at 200 GHz shown in Fig. 2c (light blue line).

The measured spectra of antennas with various $L_{ant}$ (40–200 μm) showcase a tailorable THz emission between 180 and 680 GHz. The peak of the emission shifts toward higher frequencies for the shorter $L_{ant}$ (Fig. 2c). We choose $l_{gap}$ to correspond to only a fraction of the THz resonant frequency ($l_{gap} = \frac{L_{ant}}{2}$) for all antennas. The frequency generated by optical rectification is limited by the pulse duration of the pump signal which is dispersed by the input fiber to the TFLN chip (setup described in the Methods section "Optical setup" and in detail in Supplementary Note 5A, the influence of dispersion and potential nonlinear effects inside the fiber and TFLN waveguides is discussed in Supplementary Note 3A). In combination with the lower emission efficiency of smaller bow-tie antennas the signal strength decreases for higher frequencies. Thereby our experimental results are in full agreement with finite element method simulations performed using CST Microwave studio (details in Supplementary Note 4A).

The generation length $l_{gap}$ influences the amplitude of the THz field through the effective generation length and must generally satisfy $l_{gap} < l_{coh}$. We demonstrate this dependence experimentally and fix the

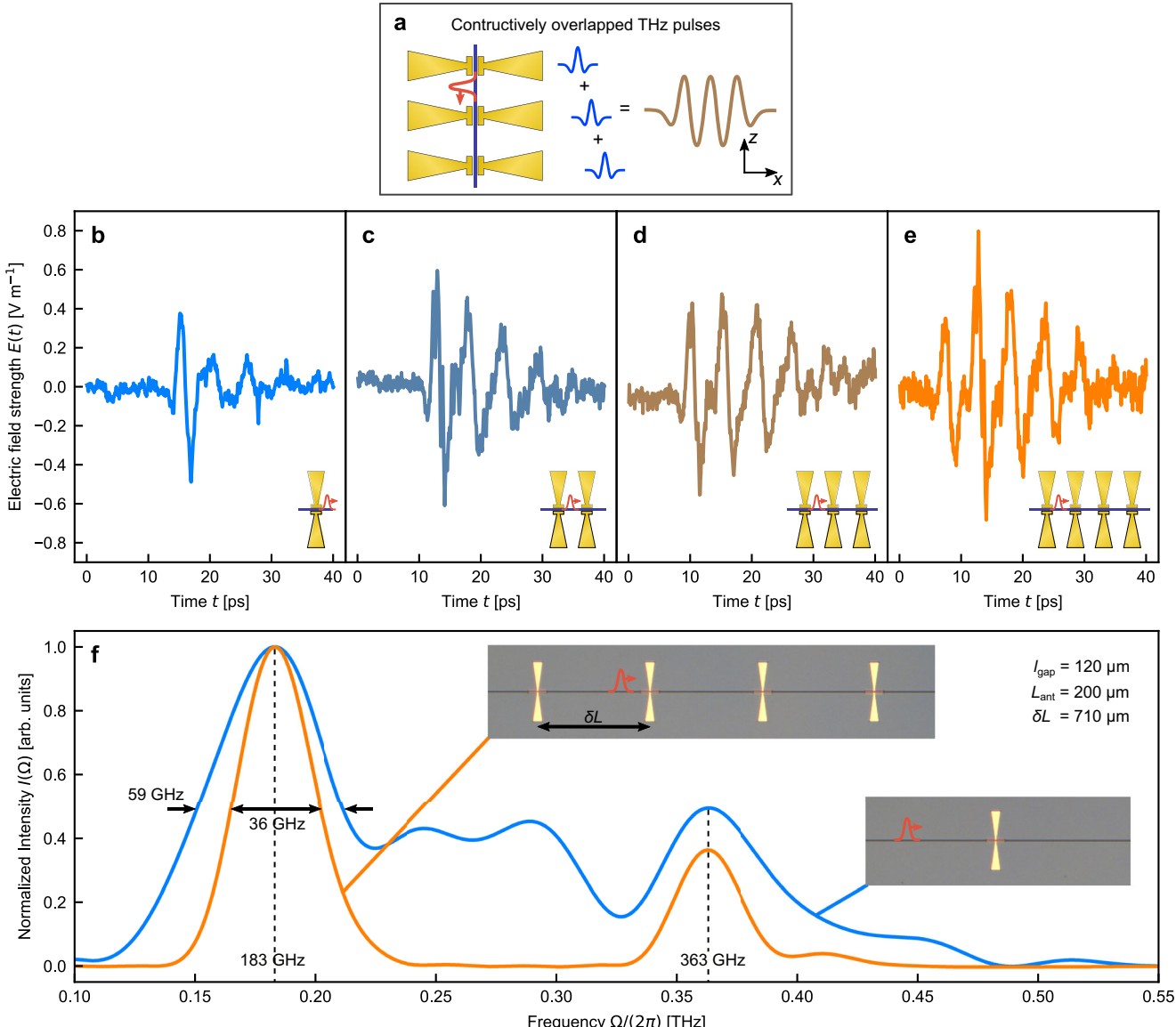

**Fig. 3 | The serial block: single-THz-cycle temporal coherence synthesis.**
**a** Schematic of the proposed serial configuration. The input pulse pumps all antennas subsequently. The synthesized THz waveform results from the spatiotemporal superposition of all emitted single-cycle pulses enabled by low-propagation loss of the TFLN platform. This allows engineering the temporal coherence at the level of one single THz cycle. **b–e** Time-domain electric field emitted by each serial block of antennas with an antenna arm length $L_{ant} = 200\,\mu m$ and a gap length of $l_{gap} = 120\,\mu m$. The distance between the antennas measures $\delta L = 710\,\mu m$ in all cases leading to a phase-shift of $2\pi$ and constructive interference of the THz pulses in the farfield. **f** Spectral intensity of 4 subsequent antennas compared to a single one. Resonance narrowing of the emitted THz frequency at 183 GHz from a full-width half maximum of 59 GHz to 36 GHz was observed. The device with 4-antennas exhibits THz emission suppression between the fundamental at 183 GHz and the first harmonic at 363 GHz due to additional periodic phase coherence condition imposed by the linear array. Both spectra are normalized to their respective peak intensity. The insets show optical microscope pictures of the serial block devices.

arm length at $L_{ant} = 90\,\mu m$ while sweeping $l_{gap}$ between 15 and 60 μm, in increments of 15 μm (Fig. 2d). The measured peak of the THz spectrum shifts toward lower frequencies for larger $l_{gap}$ due to different resonance frequencies (details in Supplementary Note 4B). Optical rectification efficiency increases with the generation length, as expected, since all generation lengths $l_{gap}$ are shorter than corresponding coherence lengths; therefore, the observed increase in the emission amplitude is not limited by phase-matching. The deviation from a quadratic increase of the intensity is well reproduced by taking into account that the resonance frequency decreases for shorter gap lengths (Fig. 2e, details about the simulations are provided in Supplementary Note 4B).

In a second experiment, we demonstrate the potential of distributed pulse phase matching to achieve full control over the

temporal coherence of the emitted THz waveform at the level of single cycles of oscillation using a serial block. We largely benefit from our ultralow-loss TFLN platform. We fabricate four devices consisting of arrays of one to four antennas ($\Omega_{res} \sim 180$ GHz) and record their emission in the timedomain (Fig. 3b–e). We keep $l_{gap} = 120\,\mu m$ below the coherence length for a frequency of 180 GHz ($l_{coh} = 178\,\mu m$). The pump pulse propagates through the antennas with the group velocity $v_g = \frac{c_0}{n_g}$ ($n_g = 2.33$) and triggers them consecutively. The spacing between antennas ($\delta L$) sets the time delay between the generated THz signals and thereby their resulting superposition in the farfield. Here, a $\delta L = 710\,\mu m$ between two successive antennas corresponds to a group delay that matches one single THz cycle at its peak resonance, resulting in a phase-matched emission at a normal exit angle from the chip. In this way, we generate few-cycle waveforms with an arbitrary

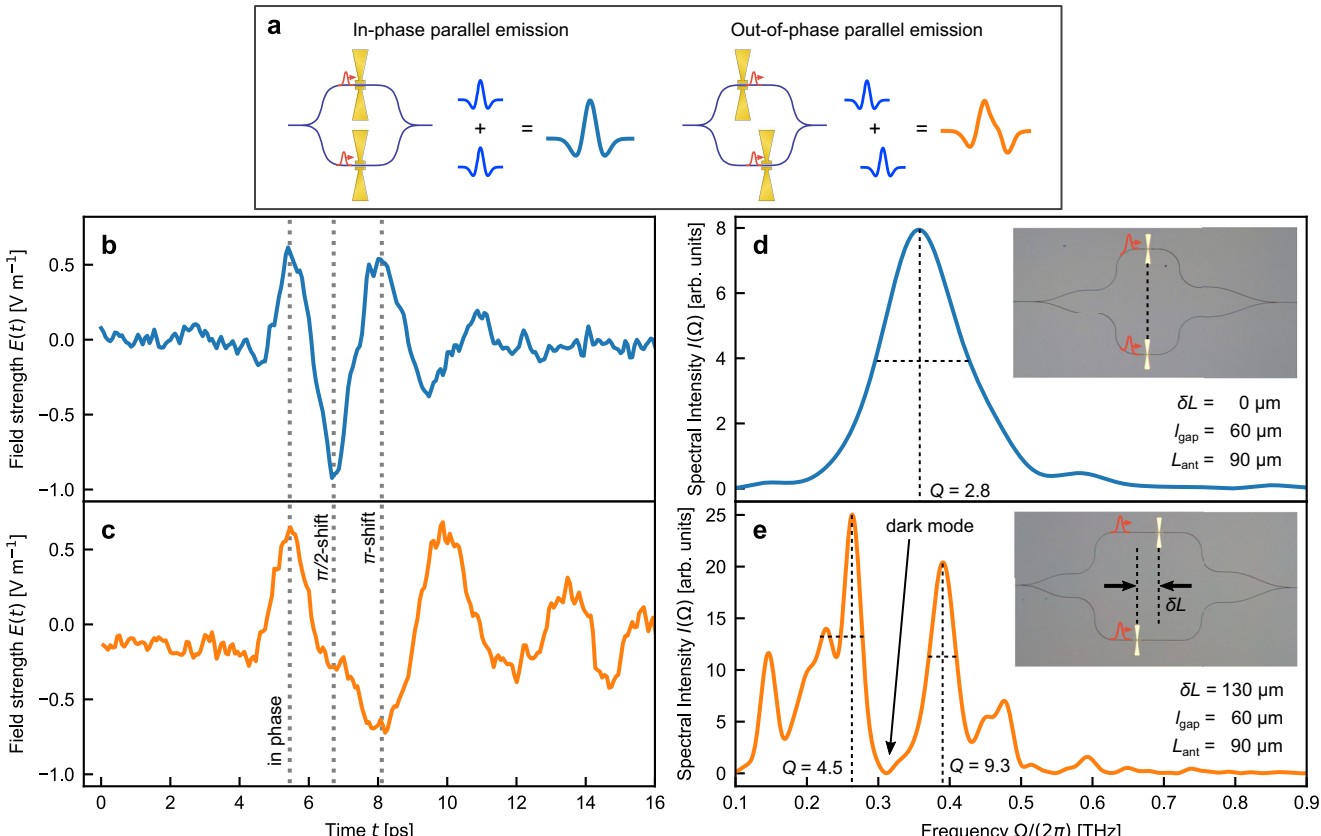

**Fig. 4 | The parallel block: sub-THz-cycle temporal phase synthesis. a** Schematic of the proposed parallel configuration. The on-chip pumping pulses are split into several parallel channels using on-chip y-splitters. Antenna's relative displacement $\delta L$ sets the phase relationship between the emitted THz pulses out-of-plane (in-phase or out-of-phase). The interference of these pulses determines the temporal shape of the farfield. **b, c** Temporal electric field evolution emitted by two in-phase antennas ($\delta L = 0\,\mu m$) and by two out-of-phase antennas ($\delta L = 130\,\mu m$, corresponding to half a THz cycle $\delta L = \frac{c_0}{n_g}\frac{T_{THz}}{2}$ or a phase delay of $\pi$). All antennas have an antenna length of $L_{ant} = 90\,\mu m$ and a gap length of $l_{gap} = 60\,\mu m$. The out-of-phase configuration displays different temporal phase properties than the in-phase configuration: at 5.5 ps, the phase of the two waveforms is identical, at 6.7 ps the phase

difference is $\pi/2$ and at 8.1 ps the phase difference is $\pi$. As highlighted by the vertical dashed lines, this phase engineering occurs on sub-THz cycle scales of the time-trace. **d, e** Spectral intensities corresponding to the time-traces shown in plot **a** and **b** are retrieved by Fourier transform. For the in-phase emission, one broad peak with a quality factor of $Q = \frac{\Omega_{peak}}{\Delta\Omega} = 2.8$ occurs. $\Delta\Omega$ is defined as the full width at half maximum spectral intensity around the particular peak angular frequency $\omega_{peak}$. For the out-of-phase emission, the phase-delay of $\pi$ at the resonance frequency leads to a suppression of the amplitude and two narrow peaks of higher quality factors $Q = 4.5$ and $Q = 9.3$. The insets show optical microscope images of the in-phase and out-of-phase devices.

number of oscillation cycles and modified temporal coherence, as shown in Fig. 3b–e. The device with four serial antenna exhibits a ~40% narrower linewidth of the spectral intensity around the fundamental resonance frequency (Fig. 3f). The second peak in the spectrum ($n = 2$, 360 GHz) corresponds to the second harmonic, since the serial block is phase matched for any $n$th harmonic of the fundamental THz frequency (due to a phase delay of $2\pi n$), and thereby also for the second harmonic.

Finally, we demonstrate the ability to engineer the temporal phase characteristics (a phase-shift of $\pi/2$ and a phase flip of $\pi$) of a waveform on sub-THz-cycle scales by a parallel block in Fig. 4. In the frequency domain, this opens up the possibility to synthesize narrow band radiation characterized with quality factors that cannot be achieved by the single bow-tie antennas. Two pumping pulses are split to individually trigger two antennas. The difference in their arrival at the antennas determines the time delay between the two emitted THz signals. We analyze two scenarios that correspond to in-phase and out-of-phase emission of THz pulses (Fig. 4a). We trigger the two antennas simultaneously (zero relative group delay) and then with a $\pi$ phase difference. We demonstrate that out-of-phase time-traces exhibit a phase shift of $\pi/2$ after half a THz cycle and a phase flip of $\pi$ after one full cycle compared to the in-phase scenario (Fig. 4b, c). For the in-phase configuration, the Fourier transform of the time-domain signal

reveals a bright mode around the resonance frequency (Fig. 4d). On the other hand, the phase change for the out-of-phase geometry manifests itself as a suppression in the emission leading to the creation of a dark mode in the farfield at 320 GHz (Fig. 4e). This allows us to achieve narrow-band radiation around 250 and 390 GHz, with quality factors of $Q = 4.5$ and $Q = 9.3$, respectively. Compared to the bare emission of bow-tie antennas which has a $Q = 2.8$, this demonstrates our ability to also synthesize narrow-band radiation with this approach.

## Discussion
In summary, we demonstrated that designer low-loss integrated circuits on TFLN platform provide unique opportunities for the synthesis of at-will THz waveforms using optical rectification. The combination of existing integrated components together with our proposed antenna scheme enables unprecedented flexibility to take control over almost all degrees of freedom of generated THz pulses from one single platform. We engineer their temporal, spectral, phase, coherence and farfield properties by various basic blocks. This could open up opportunities in the realization of high-performance and versatile THz synthesizers. Similar systems may benefit in the future from the high intrinsic electro-optic bandwidth, and fast switching of TFLN devices to offer dynamic reconfigurability of the THz emission at speeds up to

Gigahertz. Engineering the antenna geometry will allow further control of the THz properties, e.g., the generation of different polarization states (corresponding simulations are provided in Supplementary Note 4C). While the current intensity of THz pulses is limited by the pump pulse energy, we provide an extensive discussion in Supplementary Note 7 that showcases possible approaches to increase the magnitude of the emitted terahertz field. By numerically solving the nonlinear Schrödinger equation, we find that the pump pulse energy may be increased up to 100 nJ without significantly suffering from self-phase modulation effects at generated frequencies below 500 GHz (Supplementary Note 3B). Also, the control of the exact waveguide dispersion by the geometry of the waveguides may be explored to achieve pulse compression and Fourier-limited pump pulses locally in the generation region, potentially increasing the generation efficiency and the spectral bandwidth of the signal generated by optical rectification up to several THz[46]. The concept of distributed pulse phase matching may be further explored in combination with periodic poling[47–50] and group delay engineering to synthesize arbitrary patterns of THz radiation by decomposition into temporal modes[51]. Here, pump pulses at other frequencies than the telecom may also be considered toward improved confinement of the probe and terahertz beam, e.g., at wavelengths of titanium sapphire lasers. Finally, we note that LN circuits are ideal candidates also for THz metrology[12,52] where femtosecond pulses in the near-infrared could enable a sub-cycle resolution of the electric field of THz signals and have already proven resourceful for quantum applications by characterization of quantum states of light through their sub-cycle electric field signatures[53,54]. As a result, fully integrated detectors may be added to our platform. With the compatibility of chip-scale high-power lasers with TFLN platform[55], hand-held, compact time-domain spectroscopy systems may be envisioned for the future[56].

## Methods

### Fabrication

Devices are fabricated on 600-nm x-cut Lithium Niobate (LN) bonded on 2 μm of thermally grown oxide on a double-side polished silicon carrier. The waveguides are defined using negative-tone Hydrogen silsesquioxane (HSQ) (FOx-16) by means of Electron-beam lithography under multipass exposure (Elionix F-125). The pattern is then transferred to LN through a physical etching process using Reactive Ion Etching with Ar$^+$ ions. The etch depth target is 300 nm to form a ridge waveguide. We further confirm the thickness of the remaining film by means of an optical profiler (Filmetrics). We used a wet chemical process to clean the resist and redeposited material after the physical etching. The chip is annealed, then cladded with 800 nm of Inductively Coupled Vapor Deposition (ICPCVD) SiO$_2$ followed by a final annealing step[57]. We then defined the electrodes using a positive-tone poly-methyl methacrylate (PMMA) A9 series (ELS-HS50). To avoid any misalignment, the same resist was used to etch the ICPCVD SiO$_2$, and the metallic antennas. A bilayer of Ti/Au (15/285-nm) was deposited on our devices using Electron-beam evaporation (Denton).

### Optical setup

A novel dual-wavelength terahertz time-domain setup was developed to characterize the emission properties of our chip-scale emitters. A detailed sketch of the characterization setup is provided in Supplementary Note 5A. In short, femtosecond pulses from an erbium-doped fiber laser in the near-infrared are coupled from free space to the chip by grating couplers. The femtosecond pulses act as a pump signal for the THz emitters, and the emitted terahertz signal is collected from the back side (silicon side) of the TFLN chip. Two parabolic mirrors collimate and focus the emitted THz radiation onto a zinc telluride where electro-optic sampling enables the measurement of the entire temporal evolution of the emitted THz field. The probe signal at 780 nm originates from the same laser oscillator as the pump signal at

1560 nm, allowing for coherent detection of the emitted radiation. The 780 nm probing pulse overlaps spatially with the focus of the THz beam inside a nonlinear zinc telluride crystal. The interaction between THz and probe field results in a polarization change of the probe beam, which is measured in a balanced detection scheme.

## Disclaimer

The views, opinions and/or findings expressed are those of the author and should not be interpreted as representing the official views or policies of the Department of Defense or the U.S. Government.

## Data availability

The data generated in this study have been deposited in the Zenodo database under https://doi.org/10.5281/zenodo.7323903.

## Code availability

The code for plotting the data within this paper is available in the Zenodo database under https://doi.org/10.5281/zenodo.7323903.

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

## Acknowledgements

We acknowledge support from Christian Reimer from Hyperlight during fabrication and mask design and Mathieu Bertrand for fruitful discussions. A.H. acknowledges financial support from Swiss National Science Foundation (SNF) (grant 200020_192330/1), F.F.S. from the National Centre of Competence in Research Quantum Science and Technology (QSIT) (grant 51NF40-185902), A.S.-A. and M.L. acknowledge funding from Defense Advanced Research Projects Agency (DARPALUMOS) (HR0011-20-C-0137). H.K.W. acknowledges financial support from the NSF GRFP under award no. DGE1745303. I.-C.B.-C. acknowledges financial support from the Hans Eggenberger foundation (independent research grant 2019) and from the Swiss National Science Foundation (PRIMA grant PR00P2-201547). The fabrication of these chips was performed in part at the Center for Nanoscale Systems (CNS), a member of the National Nanotechnology Coordinated Infrastructure Network (NNCI), which is supported by the National Science Foundation under NSF Award no. 1541959. I.-C.B.-C. acknowledges Zoe Camille Bonzon, who provided invaluable support during the writing of the manuscript.

## Author contributions

I.-C.B.-C. designed the project and conceived the concept of THz waveform synthesis by distributed pulse phase matching. All authors developed the concept further. A.H. built the dual-color THz time-domain fiber-coupled setup with help from F.F.S. A.H., A.S.-A. and

H.K.W. carried out the measurements and acquired the data. A.H. and I.-C.B.-C. performed the CST simulations. A.S.-A. designed and fabricated the devices. A.S.-A and H.K.W. performed waveguide loss characterization by transmission spectroscopy. A.H., I.-C.B.-C., and A.S.A wrote the manuscript with help from other co-authors. The work was done under the supervision of I.-C.B.-C., M.L. and J.F.

## Competing interests

The authors declare no competing interests.
