## [Peer Review File · Nature Communications]

Terahertz waveform synthesis in integrated thin-film lithium niobate platformREVIEWER COMMENTS

Reviewer #1 (Remarks to the Author):

The authors propose a miniaturized platform for THz wave generation and flexible waveform synthesis based on thin-film lithium niobate waveguide circuits and THz antennas. By properly tailoring the dimensions of the antenna and the distributed phase matching among different antennas, the authors demonstrate versatile control of the THz temporal waveform, spectrum, amplitude, and phase. Benefiting from the low propagation loss of the pump light and low absorption of the THz radiation, one can engineer the THz waveform in a single cycle in a coherent way. The TFNL waveguide and antennas arranged in series and in parallel are studied to enhance or suppress THz emission at specific frequencies. The study here offers multidimensional control of the THz radiation, which is a significant advance in the THz source field. Here are some issues to be clarified to improve the study.

1. The pump wavelength here is 1560 nm. Please explain the reason of choosing this wavelength instead of 780 nm. Is there a challenge to prepare the LN waveguide for single mode operation at 780 nm?
2. Besides of amplitude, phase and spectrum, polarization is another key feature of the THz radiation. The reported result shows a fixed polarization state. It would be instructive if the authors add some discussion about the control of THz polarization through proper combination of the waveguide and the antennas.
3. For THz generation from bulky LN crystal, the temperature affects the THz radiation. What is the temperature effect in the waveguide circuit?
4. The generated THz pulse energy is in the order of 10-20 J, while the advantage of bulky LN is generation of strong THz radiation in the order of μJ to mJ. If one continuously increases the pump power, is there a saturation of the THz pulse energy? The effective interaction distance between the THz radiation and the pump pulse is short. There is no tilted pulse front usually used in the bulky crystal. So are there other solutions to further boost the efficiency of THz radiation, such as integrating more antennas in a waveguide?
5. The coupling of the pump pulse from the fiber to the LN waveguide affect its spectrum. The waveguide dispersion will also affect its spectrum. The variation of the pump spectrum due to the above reasons on the THz radiation should be discussed.

Reviewer #2 (Remarks to the Author):

The authors of the submitted manuscript demonstrate that a structure combining an LN waveguide and a metal antenna provides a well-controlled terahertz generation source. Furthermore, they show that by properly positioning this source in the plane of the optical circuit, the waveform of the emitted terahertz pulses can be controlled.

The proposed method is interesting as a new way to generate terahertz pulses. However, it does not appear to be impactful enough to meet the high standards of Nature Communication in the following two respects.

1. Since several methods for terahertz pulse shaping have already been proposed, it is not clear what the advantages of this method are compared to those methods. In line 17, the author states, "Currently, technologies that allow this level of refinement are entirely missing as the THz range is generally challenging to access both using optical and electronic technologies." However, the following important studies have already been reported, and the authors should clearly explain the superiority of the present method compared to these studies.

J. R. Danielson et al. "Generation of arbitrary terahertz wave forms in fanned-out periodically poled lithium niobate." *Appl. Phys. Lett.* 89, 211118 (2006).

M. Sato et al. "Terahertz polarization pulse shaping with arbitrary field control." *Nat. Photon.* 7, 724–731 (2013).

L. Gingras and D. Cooke "Direct temporal shaping of terahertz light pulses." *Optica* 4, 1416–1420 (2017).

S. Keren-Zur et al. "Generation of spatiotemporally tailored terahertz wavepackets by nonlinear metasurfaces." *Nat. Commun.* 10, 1778 (2019).

V. Muhammed "Terahertz pulse shaping using diffractive surfaces." *Nat. Commun.* 12, 37 (2021)

In particular, the number of antenna structures that can be placed in the plane is limited in this method, so it should be difficult to generate arbitrary waveforms, and this problem is difficult to overcome.

2. As the author admits in line 163, the generation efficiency of this method is only 10^{-10} (as described on page 14 of the supplement), which is extremely low compared to common THz generation methods. This is a fatal problem when used as a light source. In line 163, the author attributes this to the low excitation pulse energy. However, even if the pulse energy of the excitation is increased, the generation efficiency does not necessarily increase due to pulse spreading caused by nonlinear propagation in the LN waveguide. The strong excitation also increases the possibility of destruction of the antenna

structure. Therefore, the claim that the generation efficiency increases with an increase in the intensity of the excitation light is not convincing.

For these reasons, I conclude that the content of this manuscript does not have the high impact to be published in Nature Communication.

Reviewer #3 (Remarks to the Author):

This paper describes a new approach to the generation of THz waveforms using optical rectification in on-chip thin-film lithium niobate (TFLN) "circuits." short optical pulses are coupled into waveguides fabricated on the chip, and interact with the LN in the source region of bowtie antennas to generate the THz. The arrangement of the antennas on the chip enables one to synthesize various waveforms, including "narrow-band" multicycle pulses, pulses with designed delays or phase shifts, etc.

The technological approach presented here is novel, the discussion is rigorous, and the experiments are well performed. The paper is nicely written and well organized. I believe that the threshold for significance required for Nature Communications is well met, and as such it meets the technical and quality standards to be accepted for publication.

I do believe, however, that the paper STRONGLY overstates the originality and capability of the technical approach relative to prior published generation methods. Specifically, the authors state that "there is a very limited leverage over tailoring the generated THz emission unless through complex pump pulse shaping, external switches or multi-pulse setups" and "the phase matching properties are fixed by the refractive index mismatch at THz and pumping frequencies in bulk crystals and can not be controlled." They also state "our proposed antenna schemes enable unprecedented flexibility to take control over almost all degrees of freedom of generated THz pulses."

While indeed the author's approach addresses these issues, there is substantial prior technology that in fact solved ALL these problems, enabling much higher pulse complexity than the authors present here, and with much higher generated THz power. Approaches using periodically poled crystals to achieve quasi-phase matching or arbitrary waveform synthesis were developed in the early 2000's. Specifically, the initial demonstration in PPLN was given in

Y.-S. Lee, et al., "Generation of Narrow-Band Terahertz Radiation via Optical Rectification in Periodically-Poled Lithium Niobate," *Appl. Phys. Lett.* 76, 2505 (2000).

Cooling of the PPLN yielded narrow bandwidths (many more cycles in the waveform) than are presented in the present paper:

Y.-S. Lee, et al., "Temperature Dependence of Narrow-Band Terahertz Generation from Periodically-Poled Lithium Niobate," *Appl. Phys. Lett.* 77, 1244 (2000).

Easy tunability was demonstrated in:

Y.-S. Lee, et al., "Tunable Narrow-Band Terahertz Generation from Periodically-Poled Lithium Niobate," *Appl. Phys. Lett.* 78, 3583 (2001).

Pulse shaping was given in:

Y.S. Lee and T.B. Norris, "Terahertz Pulse Shaping and Optimal Waveform Generation in Poled Ferroelectric Crystals," *J. Opt. Soc. Am B.* 19, 2791 (2002).

Highly complex waveforms (up to 53 bits, over an order of magnitude more than the 4 bits presented here), suitable for pulse coding and other sophisticated schemes, were developed in

T. Buma and T.B. Norris, "Coded Excitation of Broadband Terahertz using Optical Rectification in Poled Lithium Niobate," *Appl. Phys. Lett.* 87, 251105 (2005).

Finally, as it is well recognized that LN has important issues due to residual THz absorption, this approach was extended to the orientation-patterned GaAs system by the Vodopyanov and Fejer group at Stanford; see e.g.

G. Imeshev, M. E. Fermann, K. L. Vodopyanov, M. M. Fejer, X. Yu, J. S. Harris, D. Bliss, and C. Lynch, "High-power source of THz radiation based on orientation-patterned GaAs pumped by a fiber laser," *Opt. Express* 14, 4439-4444 (2006).

The latter paper reported microwatt-level power generation. Higher power has been subsequently demonstrated.

So, while the present paper presents an approach having many important advantages such as on-chip generation, surface-normal emission, control of waveform, etc., the results do not exceed or even come close to meeting the performance of much older systems based on poled crystals, so the authors' claims quoted above regarding novelty and the prior difficulties faced by THz generation simply do not hold up. This needs to be rectified (pardon the pun). The paper should state the approach and results and the advantages an on-chip approach will give, but the highly overstated claims of novelty for arbitrary waveform generation, narrow-band pulses, and phase control need to be toned down.

Finally, two small changes I believe should be incorporated.

1. The emitted THz power should be directly in the abstract. THz generation technology has progressed significantly, and this critical information should be right up front in the abstract.

2. Maybe it is in the text and I just missed it, but I think the crystal orientation of the LN on the chip should be clearly stated in the main text.

Response to referees

We would like to thank all reviewers for their positive assessment of our work. In detail, we would like to thank the referee 1 for judging our work as being a significant advance in the field, referee 2 for qualifying our work as an interesting new way to generate terahertz pulses compared to other prior work and referee 3 for underlining the novelty, rigour and completeness of the presented experimental data and supporting information.

Given the suggestions and comments we received in the first round of reviews, we have decided to undertake the following major revisions to the initially submitted version of the manuscript and supplementary information:

1. We added a discussion (and corresponding simulations) to the supplementary information section D.3. regarding the generation of orthogonal and elliptical polarization states other than the the experimentally demonstrated linear polarization (Comment 3, referee 1)
2. We included a discussion (and corresponding numerical calculations) in the supplementary information section C.1 and C.2. showing the effect of self-phase modulation inside the lithium niobate waveguides at higher pump pulse powers in addition to what we experimentally illustrated in the original version (Comment 6, referee 1 and Comment 1, referee 2).
3. We extended our comparison with state-of-the-art literature in the introduction, conclusion of the main text and supplementary information section F. with respect to the strengths and unique advantages provided by our platform: cross-compatibility with photonic integrated circuits, compactness, simultaneous engineering of several properties of the generated waveform such as amplitude, frequency and polarization through chip-scale design alone (Comment 1, referee 2 and Comment 1, referee 3)
4. We discuss possible strategies that may be adopted to increase the emitted terahertz field amplitude in the supplementary information section C.2. and G. (Comment 5 and 6 referee 1, Comment 3 referee 2)
5. We have decided to slightly change the title which now reads *Terahertz waveform synthesis in integrated thin-film lithium niobate platform*. This title change now reflects better the novelty of our work with respect to the important prior works done in bulk lithium niobate.

Furthermore, we made the following small revisions to the manuscript:

1. Corrected the units of the y-axis in Fig. 1 of the supplementary information which now reads dB/cm.

Questions are shown in blue, answers are shown in black and changes to the manuscript are shown in red.

1 Reviewer #1

1. The authors propose a miniaturized platform for THz wave generation and flexible waveform synthesis based on thin-film lithium niobate waveguide circuits and THz antennas. By properly tailoring the dimensions of the antenna and the distributed phase matching among different antennas, the authors demonstrate versatile control of the THz temporal waveform, spectrum, amplitude, and phase. Benefiting from the low propagation loss of the pump light and low absorption of the THz radiation, one can engineer the THz waveform in a single cycle in a coherent way. The TFNL waveguide and antennas arranged in series and in parallel are studied to enhance or suppress THz emission at specific frequencies. The study here offers multidimensional control of the THz radiation, which is a significant advance in the THz source field. Here are some issues to be clarified to improve the study.

We thank the referee for pointing out the importance of our work.

2. The pump wavelength here is 1560 nm. Please explain the reason of choosing this wavelength instead of 780 nm. Is there a challenge to prepare the LN waveguide for single mode operation at 780 nm?

We thank the referee for pointing out the fact that our platform is - in principle - compatible with pump pulses at 780 nm as well (and by that reasoning any intermediate centre wavelength too). Given that the focus of the current study is chip-based generation of terahertz radiation, we decided to employ a fiber-based laser source instead of e.g. a titanium sapphire laser or a frequency-doubled telecom source. This comes with two important advantages: 1. generally scattering losses scale inversely with the fourth power of wavelength ($\sim \frac{1}{\lambda^4}$), and hence the sidewall roughness of the fabricated waveguides is less detrimental at larger wavelengths and 2. the photorefractive effect is more significant in the visible region. Still, we point out that low loss waveguides can be fabricated to operate also in the visible range e.g. in ref. [1], potentially allowing an even greater variety of geometries.

To address this comment from the referee, we include in the conclusion of the manuscript: "pump pulses at other frequencies than the telecom may also be considered towards improved confinement of the probe and terahertz beam, e.g. at wavelengths of titanium sapphire lasers"

3. Besides of amplitude, phase and spectrum, polarization is another key feature of the THz radiation. The reported result shows a fixed polarization state. It would be instructive if the authors add some discussion about the control of THz polarization through proper combination of the waveguide and the antennas.

Figure 1: a: Antenna structure for THz emission perpendicular to the excitation direction. The dimensions are similar to the ordinary bowtie antenna (reported in supplementary material section D) using an antenna arm length of $L_{\text{ant}} = 90 \mu\text{m}$ and a gap length of $l_{\text{gap}} = 45 \mu\text{m}$. b: Antenna response of the structure shown in (a) along the y - and z -axis simulated with CST Microwave studio plotted in logarithmic scale. c: Spiral antenna structure for elliptically polarized THz emission. A logarithmic spiral with an inner radius $r = 10 \mu\text{m}$, phase-shift of $\delta = 90^\circ$ and progression of $\alpha = 0.35$ is placed around the antenna gap formed by two gold bars of length $l_{\text{gap}} = 25 \mu\text{m}$ defining the generation area. d: Antenna response simulated with CST Microwave studio for the log-spiral antenna shown in c along y - and z -direction and the phase difference between the two different polarization directions.

We thank the referee for pointing out this important aspect. Following this suggestion, we now include a proposal that implements controlling the polarization state of the generated THz radiation. Since the highest nonlinear coefficient of lithium niobate - and therefore the most efficient THz generation - occurs along the z -axis, we align the waveguide inside the antenna gap to the y -axis of the lithium-niobate while we control the polarization of the emitted wave through the design of the antenna alone.

Fig. 1 shows the simulation results (CST Microwave studio) for two new antenna geometries on the same material platform as in the manuscript. By departing from the standard bow-tie geometry, we show that we can control the polarization state of the emitted THz wave. In Fig.1 a an antenna with bowtie-like arms is placed parallel to the y -axis of the crystal. We choose an antenna arm length of $L_{\text{ant}} = 90 \mu\text{m}$ and a gap length of $l_{\text{gap}} = 45 \mu\text{m}$ keeping the parameters similar to the original bow-tie antenna designs. Even when keeping the THz excitation along the z -

axis, the emitted signal is mainly polarized along the y -axis (Fig. 1 b). In other words, the antenna rotates the polarisation of the outcoupled THz field. Our simulations reveal that, in this case, the field ratio between the z -polarization and the y -polarization is below 4% at the resonance frequency of 370 GHz, corresponding to an intensity extinction ratio of -28 dB. This demonstrates that y -polarised terahertz emission is achieved by such an antenna structure.

Following the same principle, log-spiral antennas (Fig 1 c) can be employed to generate elliptically polarized THz signals [2]. The log-spiral with an inner radius $r = 10 \mu\text{m}$, phase-shift of $\delta = 90^\circ$ and progression of $\alpha = 0.35$ is placed around the antenna gap formed by two gold bars of length $l_{\text{gap}} = 25 \mu\text{m}$ defining the generation area. We find that such antenna design can generate elliptically polarized light (Fig 1 d).

To address this comment, we added a new section D.3 to the supplementary material saying: "Since the highest nonlinear coefficient of lithium niobate - and therefore the most efficient THz generation - occurs along the [001]-axis, within our experimental study, all the antenna structure is always oriented to emit THz fields polarized along this axis. To expand our platform by a new degree of freedom in THz control, we investigate modified antenna designs keeping the orientation of the waveguide inside the antenna gap along the y -axis of the lithium-niobate while we control the polarization of the emitted wave through the design of the antenna alone. Fig. 1 shows the simulation results for two new antenna geometries on the same material platform as in the manuscript. By departing from the standard bow-tie geometry, we show that we can control the polarization state of the emitted THz wave. In Fig. 1 a an antenna with bowtie-like arms is placed parallel to the y -axis of the crystal. We choose an antenna arm length of $L_{\text{ant}} = 90 \mu\text{m}$ and a gap length of $l_{\text{gap}} = 45 \mu\text{m}$ keeping the parameters similar to the original bow-tie antenna designs. Even when keeping the THz excitation along the z -axis, the emitted signal is mainly polarized along the y -axis (Fig. 1 b). In other words, the antenna rotates the polarisation of the outcoupled THz field. Our simulations reveal that, in this case, the field ratio between the z -polarization and the y -polarization is below 4% at the resonance frequency of 370 GHz, corresponding to an intensity extinction ratio of -28 dB. This demonstrates that y -polarised terahertz emission is achieved by such an antenna structure.

Following the same principle, log-spiral antennas (Fig 1 c) can be employed to generate elliptically polarized THz signals [2]. The log-spiral with an inner radius $r = 10 \mu\text{m}$, phase-shift of $\delta = 90^\circ$ and progression of $\alpha = 0.35$ is placed around the antenna gap formed by two gold bars of length $l_{\text{gap}} = 25 \mu\text{m}$ defining the generation area. We find that such antenna design can generate elliptically polarized light (Fig 1 d)."

4. For THz generation from bulky LN crystal, the temperature affects the THz radiation. What is the temperature effect in the waveguide circuit?

We thank the referee for anticipating that decreasing the temperature can affect the absorption coefficient of LN in the terahertz band due to the suppression of phonons in this spectral range [3]. Such loss mechanism is dominant in bulk LN crystals since the generation takes place a few hundred micrometers below the crystal surface. As a result, to avoid a substantial decrease in the generated wave power, a cooling mechanism has to be employed [4]. In contrast to the bulk case, in the present work, the THz is generated in a 600 nm thick layer of lithium niobate (highly sub-wavelength, approx. 1/100 of the terahertz wavelength) and emitted perpendicularly to it. Such small scales minimize the interaction of the generated wave with the “THz-lossy” material. As we show from our THz-transmission studies (Supplementary material section A.2.) the propagation loss of the THz is negligible in our case in the lithium niobate layer.

Therefore, we anticipate that cooling the crystal may not bring the same benefits as it brings in the case of bulk crystals. Unfortunately, we are unable to verify this hypothesis experimentally at the moment.

To address this comment from the referee, we now add to our discussion about minimized absorption in TFLN in the supplementary material the following comment: "Given that the propagation loss of the terahertz radiation inside a thin layer of lithium niobate is negligible, we presume that cooling the chip may not lead to a significant increase in the emitted terahertz field amplitude."

5. The generated THz pulse energy is in the order of 10^{-20} J, while the advantage of bulky LN is the generation of strong THz radiation in the order of μJ to mJ . If one continuously increases the pump power, is there a saturation of the THz pulse energy? The effective interaction distance between the THz radiation and the pump pulse is short. There is no tilted pulse front usually used in the bulky crystal. So are there other solutions to further boost the efficiency of THz radiation, such as integrating more antennas in a waveguide?

We agree that the generated power of our platform is not comparable to bulk crystals at the moment, as our main focus was rather directed to demonstrate the novel capabilities of integrated platforms in contrast to bulk crystals, which are excellent sources of intense terahertz pulses.

Nevertheless, to address this comment of the referee we identify several strategies that can be followed to increase the terahertz field amplitude at the detector. This entails two different approaches that need to be tackled in tandem:

1. increasing the generated terahertz field in-device and then
2. the further optimization of its outcoupling and beamforming into the farfield such that a receiver located in the farfield received more terahertz power.

- (a) Pumping several waveguides in parallel by interfacing our devices with fiber-based lasers,
- (b) Increasing the pump pulse energy,
- (c) Engineering the waveguide dispersion [5, 6] and hence the group index of the pump pulses locally,
- (d) Improving the directivity of the terahertz beam by attaching an index-matched silicon lens on the back of the chip.

In detail, one could imagine using the available pump power on-chip more efficiently to avoid most of the pump remaining undepleted. One can use several subsequent antennas to improve efficiency. Additionally, periodic poling can be employed to increase the effective number of antennas by a factor of 2 by flipping the crystal domains. Also, the number of parallel channels may be increased as well. We calculate that an array of 20×20 antennas resonating at 360 GHz would fit onto a chip of $5 \text{ mm} \times 2.7 \text{ mm}$, thereby increasing the total field amplitude by a factor of 20. In this case, the temporal pulse spreading can be managed by tailoring the waveguide dispersion as in refs.[5, 6].

We now provide additional numerical calculations in section C.2 of the supplementary information showing that for an increase of the of the pump energy to 100 nJ self-phase modulation of the pump signal within the generation area will affect the THz generation. Consequently the emitted terahertz field amplitude can potentially be increased by a factor of 1000. By implementing 20 antennas in parallel, we project that the field amplitudes on the order of 20 kV/m should be achievable for this system. A more detailed discussion about the power limitation caused by nonlinear effects is also given in response to question 3 of reviewer #2.

Finally, the THz energy reported in the present work is the as-measured value ignoring any loss in the system. For instance, the THz energy reported is what is transmitted through ≈ 30 cm of unpurged path length, collected with parabolic mirrors and focused onto the detection crystal without any correction. By plugging in the numbers, we find that due to the high refractive index of the silicon substrate in the THz range ($n_{\text{SI}} = 3.425$ [3]), the angular width (110° according to the farfield in CST simulations, $\text{NA} = 0.82$) exceeds the parabolic mirror ($\text{NA} = 0.5$). Consequently, the measured THz pulse energy is lower than the full signal emitted by the device by an estimated factor of 2.7. A silicon lens can be attached directly into the substrate to improve the collection for the directed farfield, similar to commercial photoconductive emitters. [7, 8].

To address this comment from the referee, we now include to the conclusion of the main manuscript: "While the current intensity of THz pulses is limited by the pump pulse energy, we provide an extensive discussion in the supplementary information section G that showcases possible approaches

to increase the magnitude of the emitted terahertz field. By numerically solving the nonlinear Schrödinger equation, we find that the pump pulse energy may be increased up to 100 nJ without significantly suffering from self phase modulation effects at generated frequencies below 500 GHz (supplementary information section C 2).”

Additionally, we add a new section G to the supplementary material:

While we achieve THz fields in the range of only 1 V/m within the current study, various optimizations of the demonstrated on-chip THz source may allow the generation of stronger THz signals in the future.

First of all, one could imagine using the available pump power on-chip more efficiently to avoid most of the pump remaining undepleted. One can use several subsequent antennas to improve efficiency. Additionally, periodic poling can be employed to increase the effective number of antennas by a factor of 2 by flipping the crystal domains. Also, the number of parallel channels may be increased as well, for example by placing 20×20 or more antennas on chip. In this case, the temporal pulse spreading can be managed by tailoring the waveguide dispersion as in refs.[5, 6].

In our current investigation the pulse energy of the pump signal measures only 100 pJ. In section C 2 we show that for an increase of the of the pump energy to 100 nJ self-phase modulation of the pump signal within the generation area will affect the THz generation. Consequently the emitted terahertz field amplitude can potentially be increased by a factor of 1000. By implementing 20 antennas in parallel, we project that the field amplitudes on the order of 20 kV/m should be achievable for this system.

Finally, the THz energy reported in the present work is the as-measured value ignoring any loss in the system. For instance, the THz energy reported is what is transmitted through ≈ 30 cm of unpurged path length, collected with parabolic mirrors and focused onto the detection crystal without any correction. By plugging in the numbers, we find that due to the high refractive index of the silicon substrate in the THz range ($n_{\text{SI}} = 3.425$ [3]), the angular width (110° according to the farfield in CST simulations, $NA = 0.82$) exceeds the parabolic mirror ($NA = 0.5$). Consequently, the measured THz pulse energy is lower than the full signal emitted by the device by an estimated factor of 2.7. A silicon lens can be attached directly into the substrate to improve the collection for the directed farfield, similar to commercial photoconductive emitters. [7, 8].

In conclusion, the above improvements are expected deliver terahertz field amplitudes of about 20 kV/m, which is on-par with other electro-optic emitters used e.g. in time-domain spectroscopy, sensing.”

6. The coupling of the pump pulse from the fiber to the LN waveguide affect its spectrum. The waveguide dispersion will also affect its spectrum. The variation of the pump spectrum due to the above reasons on the THz

radiation should be discussed.

We thank the referee for pointing out this important aspect. To address

Figure 2: **Pump power coupling to TFLN chip** a: Spectral power density of the pump signal before it is coupled into the LN waveguides for the two different power configurations used in the current work. b: Spectrum of the near-infrared signal out-coupled from first-generation sample. The input power measured 64 mW. c: Spectral density of the pump pulses after the second-generation sample for 64 mW (blue) and 31 mW (orange) of input power.

this comment, we now introduce an extensive discussion and corresponding computations in section B.1. and C.1. of the supplementary material.

In short, in our experiments, we use a 1.1 m long fiber to couple the pump signal into the TFLN waveguides. The pump signal has higher pulse energy, wider spectral bandwidth and a longer propagation length inside the fiber than the TFLN waveguides (1.1 m fiber length versus 0.5-3 mm long waveguides). Consequently, in our case, spectral and temporal changes of the pump signal are dominated by the dispersion and nonlinear effects inside the fiber as well as the grating couplers, which have a bandwidth lower than the one of the femtosecond pulses. The exact spectra coupled into the chip and coupled out of the chip are shown in Fig. 2. Still, we quantify the effect of the waveguide dispersion mentioned by the referee on the terahertz amplitude below. This will be particularly critical for higher pulse energies for achieving higher THz field amplitudes.

To illustrate the influence of the pump signal on the THz properties, we investigate the THz spectrum generated by a Gaussian pump pulse of different lengths. The signal generated in the optical rectification process is mainly determined by the shape of the envelope of the pump pulse $I(t)$. If we neglect the influence of phase-matching, the spectral electric field generated inside the antenna gap is determined by the term:

$$E_{\text{THz}}(\Omega) \sim I(\Omega) \cdot \Omega, \quad (1)$$

with the angular frequency of the THz (Ω) and the Fourier transformation of the incoming intensity envelop $I(\Omega)$. For a Gaussian pulse with a pulse length $\tau = t_{\text{FWHM}}/(2\sqrt{2\ln 2})$, pulse energy E_{pulse} and effective mode size

Figure 3: a: Calculated THz spectrum generated by a Gaussian pump pulse of varying pulse length, but constant pulse energy. b: THz spectrum calculated from the measured pump spectra shown in the Fig. 2 considering the pump signal being Fourier-limited at the antenna (solid line) or chirped by the dispersion of the optical fiber and the TFLN waveguide (dashed line).

A_{mode} , $I(\Omega)$ is given by

$$I(\Omega) = \frac{E_{\text{pulse}}}{A_{\text{mode}}\sqrt{2\pi}} \exp\left(-\frac{\tau^2\Omega^2}{2}\right). \quad (2)$$

As shown in Fig. 3 a, E_{THz} increases linearly with frequency in the low spectral range independently from the pump pulse length. The latter determines the upper-frequency limit. With increasing pulse length the deviation from the linear increase shifts towards lower frequencies. The slope of the increase is determined by the pulse energy E_{pulse} per mode size A_{mode} .

To understand how the spectral changes inside the fiber and waveguide can affect the optical rectification process, we now calculate the generated THz spectrum based on the pump spectra presented in Fig. 2 for two simplified cases. First, assuming the pump pulses are Fourier-limited (solid line in figure 3 b) and secondly including a group delay dispersion of -0.02 ps^2 (dashed line in figure 3 b) corresponding to the propagation in 1.1 m of optical fiber ($\text{GVD} = -19.65 \text{ ps}^2/\text{km}$) and 5 mm of TFLN waveguide ($\text{GVD} = 280 \text{ ps}^2/\text{km}$, determined for particular waveguide design with COMSOL Multiphysics Software). The phase shift caused by nonlinear effects inside the optical fiber and waveguide strongly depends on the pulse energy but will counteract the negative group delay dispersion. Consequently, the true pulse length will be between those two simplified cases.

Below 400 GHz the calculated spectral THz field is barely influenced by the different spectra or the considered chirp (Figure 3 b). For higher frequencies we observe a flattening of the THz field generated by the chirped

pulses, while the spectral differences still do not influence the efficiency of the THz generation in a crucial manner.

In our specific experimental conditions, most of the investigated devices emit frequencies below 400 GHz. Therefore, the dispersion and nonlinear effects do not influence the THz signal significantly. For the study of the different antenna sizes, the spectral and temporal changes inside the fiber and TFLN waveguide have to be taken into account since we reach frequencies up to 680 GHz. For this study, we decreased the pump power to exclude the influence of nonlinear effects in both - the fiber and waveguide - on our measurement results.

To address this comment from the referee, we now include a general discussion about the influence of the pump pulse shape onto the optical rectification to section B.1:

"For small generation lengths below the coherence length $l_{\text{coh}} = \frac{c}{4n_g f_{\text{THz}}}$ the spectral properties of the generated field are mainly determined by the pulse length t_{FWHM} of the near infrared signal. For a Gaussian pulse with a pulse length $t_{\text{FWHM}} = 2\sqrt{2 \ln 2} \tau$, pulse energy E_{pulse} and effective mode size A_{mode} , $I(\Omega)$ is given by

$$I(\Omega) = \frac{E_{\text{pulse}}}{A_{\text{mode}}\sqrt{2\pi}} \exp\left(-\frac{\tau^2\Omega^2}{2}\right). \quad (3)$$

As shown in Fig. 3 a, E_{THz} increases linearly with frequency in the low spectral range independently from the pump pulse length. The latter determines the upper frequency limit. With increasing pulse length the deviation from the linear increase shifts towards lower frequencies. The slope of the increase is determined by the pulse energy E_{pulse} per mode size A_{mode} ."

Additionally, we inserted a new section C.1. to the supplementary information discussing these influences for our particular measuring configurations:

"In section B 1 we have shown the THz spectrum generated by optical rectification depends on the pulse shape of the NIR pump signal. To understand how the spectral and temporal changes inside the fiber and waveguide can affect the optical rectification process we now calculate the generated THz spectrum based on the pump spectra presented in the supplementary material figure 10 b and c for two simplified cases. First assuming the pump pulses being Fourier-limited (solid line in figure 3 b) and secondly including a group delay dispersion of -0.02 ps^2 (dashed line in figure 3 b) corresponding to the propagation in 1.1 m of optical fiber ($GVD = -19.65 \text{ ps}^2/\text{km}$) and 5 mm of TFLN waveguide ($GVD = 280 \text{ ps}^2/\text{km}$, determined for particular waveguide design with COMSOL Multiphysics Software). The phase shift caused by nonlinear effects inside the optical fiber and waveguide strongly depend on the pulse energy but will counteract the negative group delay

dispersion. As a consequent, the true pulse length will be in between those two simplified cases.

Below 400 GHz the calculated spectral THz field is barely influenced by the different spectra or the considered chirp (Figure 3). For higher frequencies we observe a flattening of the THz field generated by the chirped pulses, while the spectral differences still do not influence the efficiency of the THz generation in a crucial manner.

In our specific experimental conditions, most of the investigated devices emit frequencies below 400 GHz. Therefore, the dispersion and nonlinear effects do not influence the THz signal significantly. For the study of the different antenna sizes, the spectral and temporal changes inside the fiber and TFLN waveguide have to be taken into account, since we reach frequencies up to 680 GHz. For this study we decreased the pump power to exclude the influence of nonlinear effects in both - the fiber and waveguide - onto our measurement results. For simulations we are comparing to the measured results, nonlinear effects will not be taken into account.”

2 Reviewer #2

1. The authors of the submitted manuscript demonstrate that a structure combining an LN waveguide and a metal antenna provides a well-controlled terahertz generation source. Furthermore, they show that by properly positioning this source in the plane of the optical circuit, the waveform of the emitted terahertz pulses can be controlled. The proposed method is interesting as a new way to generate terahertz pulses. However, it does not appear to be impactful enough to meet the high standards of Nature Communication in the following respects. Since several methods for terahertz pulse shaping have already been proposed, it is not clear what the advantages of this method are compared to those methods. In line 17, the author states, "Currently, technologies that allow this level of refinement are entirely missing as the THz range is generally challenging to access both using optical and electronic technologies." However, the following important studies have already been reported, and the authors should clearly explain the superiority of the present method compared to these studies.
 - [9] J. R. Danielson et al. "Generation of arbitrary terahertz wave forms in fanned-out periodically poled lithium niobate." Appl. Phys. Lett. 89, 211118 (2006).
 - [10] M. Sato et al. "Terahertz polarization pulse shaping with arbitrary field control." Nat. Photon. 7, 724–731 (2013).
 - [11] L. Gingras and D. Cooke "Direct temporal shaping of terahertz light pulses." Optica 4, 1416–1420 (2017) .

- [12] S. Keren-Zur et al. “Generation of spatiotemporally tailored terahertz wavepackets by nonlinear metasurfaces.” *Nat. Commun.* 10, 1778 (2019).
- [13] V. Muhammed “Terahertz pulse shaping using diffractive surfaces.” *Nat. Commun.* 12, 37 (2021)

We thank the reviewer for judging that our proposed method is interesting and that it provides a new way to generate terahertz pulses. However, we respectfully disagree with assessing our work not to be impactful based on the comparison with the mentioned works. Clearly, there exist prior works which have successfully generated terahertz transients with various characteristics. But, most of the works so far however are performed in bulk nonlinear crystals (among which lithium niobate) or by shaping the terahertz waveform after its generation by pulse shaping elements positioned after the emitter.

Our approach here shapes the generated waveform at the emitter in a miniaturized chip compatible with a rich palette of integrated photonics components. This approach and the extensive experimental study that uses such low-loss, highly diverse integrated nonlinear photonic circuits for terahertz generation pumped by femtosecond pulses is to our knowledge the first demonstration in the field. We believe that this work showcases a path forward to combine a variety of elements on one single architecture to create new opportunities for THz photonics.

Nevertheless, we regret the lack of clarity from our side in the first version of the manuscript with respect to advantages of our work compared to those works. Also, we agree that the mentioned sentence is not specific enough and has been removed.

Consequently, we address this comment by providing a comparative analysis below to the works mentioned by the referee. We see clearly that (with the exception of the metasurface work), the mentioned works use bulk crystals or pulse shaping after the emitter (p.s.a.e.).

We hope that this comparison underlines the advantages that our platform provides, which are not at all, or only hardly possible with other techniques mentioned by the referee. Instead, while other works covers certain degrees of freedom in tailoring the THz radiation, our work can control all properties of the THz wave using a single platform (see table 1).

- (a) **the on-chip waveguides** provide compactness and compatibility with fiber-based source and detector technologies. This is critical since experiments using conventional Ti:Sapphire lasers suffer from huge pulse broadening due to large GVD of standard optical fibers. Also, by using available telecommunication infrastructure experiments can be done without the challenge of free-space propagation of the pump pulses,

	method	broad	narrow	polarization	spatial	at source
Our work Integrated circuits in TFLN	On-chip circuits	✓	✓	✓(sim.)	✓	✓
Quasi-phasematched crystals [14, 15, 16, 17]	bulk	✗	✓	✗	✗	✓
Fanned-out PPLN [9]	bulk	✗	✓	✗	✗	✓
Polarization shaping [10]	bulk	✓	(✓)	✓	✗	✓
Photo-injected reflection [11]	p.s.a.e.	✗	✓	✗	✗	✗
Nonlinear Metasurfaces [12]	free-space	✓	✗	(✓)	✓	✓
Diffractive surfaces [13]	p.s.a.e.	✗	✓	✓	✓	✗

Table 1: Comparison of the flexibility in the THz properties offered by integrated circuits in TFLN to common THz generation approaches. p.s.a.e: pulse shaping after the emitter

- (b) **the design of the individual antenna and the antenna distribution on-chip** provides the possibility to custom-tailor the radiation at the source - rather than through a spectral shaping element placed after a broadband emitter (which is the case for spatial light modulators [18]), thereby minimizing the effective loss,
- (c) **the low propagation loss characteristic for our fabrication process** allows for complex photonic architectures, where several antennas may be placed downstream rather than using discrete components with considerable insertion losses,
- (d) **antenna structures can also be used to pole the lithium niobate film,**
- (e) **the thermal and mechanical stability** of chip-based devices. For example, the operation point of integrated interferometers has better stability compared to hand-made free-space ones,
- (f) **the simultaneous control over several degrees of freedom** by chip-scale design alone (i.e. in our case all emitters are pumped under the same conditions, just the chip design is changed),
- (g) **the control of the pump-pulse propagation both temporally (compression or broadening) and spectrally** using waveguide geometric dispersion [5, 6]. For example, by changing the dimen-

sions of the waveguide, one can launch desired pulses with proper characteristics to the antenna where the THz generation occurs.

- (h) outlook on combining our demonstration with active chip-scale components that are intrinsically fast (e.g. GHz-speed intensity modulators to time the pump pulses, multi-port splitters to distribute pump pulses on-chip, or integrated laser sources compatible with TFLN towards fully-integrated THz sources [19]),
- (i) outlook to combine the emitters with perfectly matched detectors on the same chip.

To address this comment from the referee, we now refer to these works in the introduction of the manuscript. Furthermore, we provide in the supplementary material section F an extensive discussion of this prior work and of the advantages of our technique:

In this work, we depart from the commonly employed approach to generate terahertz transients in bulk nonlinear crystals and instead demonstrate on-chip generation that explores many of the advantages of integrated photonic circuits. Nevertheless, various prior works have attempted to generate terahertz transients with various characteristics. Since most of the works so far however are performed in bulk nonlinear crystals (among which lithium niobate) or by shaping the terahertz waveform after its generation by pulse shaping elements positioned after the emitter, our approach here using integrated circuits is an entirely different approach.

We provide in the table below a comparative analysis with some of the works that have succeeded in the past to custom-tailor the emission of terahertz transients. Approaches based on quasi-phase matching as e.g. periodically poled LN generate narrowband multi-cycle radiation, where the emission frequency is controlled by the domain period [14, 15, 16, 17, 20]. In fanned-out PPLN, the spatially varying domain period allows to control the frequencies contributing to the THz signal [9]. Designing PPLN the signal strength emitted from each individual domain is directly linked to the domain thickness. In contrast, in our case, the distance of the antennas can be varied independently from the gap length determining the signal strength giving a further degree of freedom in the waveform generation. TFLN offers the ability of poling the LN as well allowing more efficient devices by reducing the required antenna distance. While in PPLN the number of periods is limited by absorption of THz signal, the perpendicular antenna emission circumvents THz propagation through the LN. Due to the low optical losses in TFLN waveguides, the number of antennas can be further increased, mainly limited by the desired chip size. Poling of LN allow the modification of the crystallographic structure only along the [001] axis, so that the THz polarization is fixed to one direction within a PPLN crystal. The overlap of THz pulses via photo-injected reflection offers a almost similar degree of flexibility [11], but also in this approach the THz field is determined by the parallel-plate waveguide pre-

venting any polarization modification. THz shaping based on diffractive surfaces is another promising approach offering besides the spectral and temporal tuning the control over the polarization properties of the THz radiation. However, all the above mentioned approaches rely on the overlap of many single THz pulses determining spectral and temporal properties. Inherently, these approaches are not suited for the generation and control of few- or single-cycle THz pulses with broadband spectra. Using single or parallel antenna emitters, we have the additional possibility to few-cycle pulses maintaining the control of the center frequency by the antenna design. Using GaP instead of LN, the crystallographic structure allows the generation of single-cycle THz field along two perpendicular axis depending on the pump signal's polarization state. Consequently shaping the polarization of the incoming pulses by the use of an optical pulse shaper gives the full control of the THz polarization [10]. This approach might be combined with quasi-phase-matched GaP generating many-cycle waveforms combined with polarization control. Nevertheless we do not see the opportunity to also control the spatial pattern of the THz radiation, as the distribution of several antennas within the 2D-chip plane of TFLN can offer. Within a nonlinear meta-surface the variation of the antenna orientation is utilized to create custom-tailored spatial THz field pattern, but this approach is restricted to the generation of single-cycle pulses missing the possibility of arbitrary waveform generation [12].

Consequently, using integrated photonic circuits provides unique flexibility of tailoring THz radiation within a single technology and this approach shows the potential to be further developed into a fully integrated design.

As a result of this comparison, we now underline the clear unique advantages that our platform provides. While other works covers certain degrees of freedom in tailoring the THz radiation, our work can control all properties of the THz wave using a single platform (see table 1).

- (a) **the on-chip waveguides** provide compactness and compatibility with fiber-based source and detector technologies. This is critical since experiments using conventional Ti:Sapphire lasers suffer from huge pulse broadening due to large GVD of standard optical fibers. Also, by using available telecommunication infrastructure experiments can be done without the challenge of free-space propagation of the pump pulses,
- (b) **the design of the individual antenna and the antenna distribution on-chip** provides the possibility to custom-tailor the radiation at the source - rather than through a spectral shaping element placed after a broadband emitter (which is the case for spatial light modulators [18]), thereby minimizing the effective loss,
- (c) **the low propagation loss characteristic for our fabrication process** allows for complex photonic architectures, where several an-

tennas may be placed downstream rather than using discrete components with considerable insertion losses,

- (d) antenna structures can also be used to pole the lithium niobate film,*
- (e) the thermal and mechanical stability of chip-based devices. For example, the operation point of integrated interferometers has better stability compared to hand-made free-space ones,*
- (f) the simultaneous control over several degrees of freedom by chip-scale design alone (i.e. in our case all emitters are pumped under the same conditions, just the chip design is changed),*
- (g) the control of the pump-pulse propagation both temporally (compression or broadening) and spectrally using waveguide geometric dispersion [5, 6]. For example, by changing the dimensions of the waveguide, one can launch desired pulses with proper characteristics to the antenna where the THz generation occurs.*
- (h) outlook on combining our demonstration with active chip-scale components that are intrinsically fast (e.g. GHz-speed intensity modulators to time the pump pulses, multi-port splitters to distribute pump pulses on-chip, or integrated laser sources compatible with TFLN towards fully-integrated THz sources [19]),*
- (i) outlook to combine the emitters with perfectly matched detectors on the same chip.*

2. In particular, the number of antenna structures that can be placed in the plane is limited in this method, so it should be difficult to generate arbitrary waveforms, and this problem is difficult to overcome.

In our study, we show experimental results from up to 4 serial antennas and two parallel antennas. However, the low loss of TFLN allows for a larger number of parallel channels. We estimate that in future studies, by combining periodic poling (already demonstrated in TFLN [21]) with both serial and parallel waveguides, creating an array of 20×20 or more antennas on-chip is possible.

This comment of the referee is now covered by the newly added section G in the supplementary material (which is also reported in the answer to referee 1, comment 5) discussing besides possibilities to increase the generated THz signal strength also increasing the number of antennas on a chip.

3. As the author admits in line 163, the generation efficiency of this method is only 10^{-10} (as described on page 14 of the supplement), which is extremely low compared to common THz generation methods. This is a fatal problem when used as a light source. In line 163, the author attributes this to the low excitation pulse energy. However, even if the pulse energy

of the excitation is increased, the generation efficiency does not necessarily increase due to pulse spreading caused by nonlinear propagation in the LN waveguide. The strong excitation also increases the possibility of destruction of the antenna structure. Therefore, the claim that the generation efficiency increases with an increase in the intensity of the excitation light is not convincing.

We thank the referee for touching upon this important point, and regret the lack of completeness in the original version of the manuscript. We fully agree with the referee that in general nonlinear pulse spreading impacts negatively the emitted terahertz field. In detail, the nonlinear effects and the corresponding power limit depend on the spectral and temporal properties of the NIR signal at the location of the generation antennas. These are strongly related to the pump pulse energies. Already in our initially submitted version of the manuscript, we demonstrate that in our case, the dispersion of the pulses is mainly affected by the fiber and its spectrum mainly by the grating couplers used to couple to the chip, hence SPM does not play a role in the presented data. In the revised manuscript we provide a quantitative numerical analysis based on a publicly available solver of the nonlinear Schrodinger equation that calculates the self phase modulation of the pump pulses inside the lithium niobate waveguides at pump powers up to 100 nJ as well.

We use the python package fmas [22] that solves the unidirectional nonlinear Schrodinger equation to simulate the pulse propagation inside the TFLN waveguides. The total dispersion is calculated starting from the effective mode refractive index that we simulated using the COMSOL Multiphysics Software. We assume a Kerr nonlinear refractive index to be the one of bulk LN: $n_2 = 1.8 \cdot 10^{-19} \text{ m}^2/\text{W}$ [6]. Furthermore, Raman scattering is expected to occur in lithium niobate. Here, we chose values of $f_R = 0.635$ for the Raman fraction, a period time of $\tau_1 = 21 \text{ fs}$ and a decay of $\tau_2 = 544 \text{ fs}$ in our simulation [23]. We note that the different values for f_R reported in the literature cause uncertainty in our simulations.

As an example, we investigate the scenario of pumping the TFLN chip (with waveguide dimensions as in our experiments) with Fourier-limited Gaussian pulses with a FWHM intensity duration of 500 fs and 1 nJ energy (10 times higher than in our experiments). Our findings reveal that using such a pulse length still contributes to optical rectification up to 500 GHz. However, shortening the pulse (increasing the peak power), would result in strong nonlinear effects. We do not find any significant change in the temporal pulse shape up to distances of 4 mm for our given waveguide geometry. Hence, we predict the terahertz generation to remain unaffected for this length (Fig. 4). In the frequency domain we can see the spectral broadening due to nonlinear effects (Fig. 4). Further, we find that pulses of 100 nJ (1000 times higher than in our experiment) propagate without significant pulse broadening over a length of 500 μm , even if strong spec-

Figure 4: Simulation of a 500 fs-long Gaussian pulse with a pulse energy of 1 nJ in a 5 mm long TFLN waveguide performed with the python package `fmas` [22]. Left: Evolution of the temporal pulse-shape. Right: Signal evolution in the spectral domain.

tral broadening occurs (Fig. 5). From these simulations, we conclude that pulse energies in the range of ~ 100 nJ will be the upper power limit. If successfully guiding a 500 fs pulse into one of our antenna gap, the amplitude of such a single-antenna emission is expected to increase by 3 orders of magnitude to 1 kV/m and the pulse energy of the THz signal can reach ~ 10 fJ. However, photonic integrated circuits may prove beneficial. For example, one could envision pumping several waveguides in parallel (e.g. up to 20), leading to a total terahertz electric field of 20 kV/m. To further increase the pump pulse energy several strategies such as pre-chirping the pulses, increasing the mode-size in between the antennas by adiabatic tapers and wide waveguides can be implemented. Finally, further increasing the strength of the emitted signal may be achieved by attaching a silicon lens behind the chip for better directivity. Studying these vast possibilities in-depth goes beyond the scope of this work.

Regarding the damage threshold of the antenna: since the NIR mode is very well confined inside the LN waveguide resulting in a low overlap/interaction with the gold structure (see Fig. 2.b. in the supplementary material), we do not expect the antenna structures to suffer from potential damage in this range of pump pulse energies. Finally, the large band-gap of lithium niobate provides low absorption losses below 0.727 dB/cm (as shown in the supplementary material section A.1.) in the NIR, circumventing heating of the integrated structure even for increased pulse energy.

In conclusion, the above improvements are expected deliver terahertz field amplitudes of about 20 kV/m, which is on-par with other electro-optic emitters used e.g. in time-domain spectroscopy, sensing. High-field experiments will either need to resort to additional field confinement e.g. provided by antennas patterned on the chips to be investigated (which can easily reach field enhancement factors of 1000), or to bulk nonlinear

Figure 5: Simulation of a 500 fs-long Gaussian pulse with a pulse energy of 100 nJ in a 0.5 mm long TFLN waveguide performed with the python package fmas [22]. Left: Evolution of the temporal pulse-shape. Middle: Signal evolution in the spectral domain. Right: Pulse shape after the propagation length of a typical antenna gap length (150 μm) and a device of several antennas (0.5 mm) compared to the initial pulse shape.

generation crystals.

We added now to the Discussion and conclusion section of the main text: "we provide an extensive discussion in the supplementary information section G that showcases possible approaches to increase the magnitude of the emitted terahertz field. By numerically solving the nonlinear Schrödinger equation, we find that the pump pulse energy may be increased up to 100 nJ without significantly suffering from self phase modulation effects at generated frequencies below 500 GHz (supplementary information section C 2)." Additionally, we included two new sections to the supplementary material discussing the limitations of the pump power due to nonlinear effects based on these simulations (section C 2) and the THz field strength potentially achievable by the mentioned improvements of our technology (section G). We provide here the content of section C 2:

"While for the current study comparably low pump pulse energy of up to 100 pJ are utilized, the question arises as to how much we can increase the pump power in the future - especially to improve the generation efficiency and the emitted THz intensity. The nonlinear effects occurring inside the fiber coupling the laser signal to the TFLN chip can in principle be mitigated using free-space propagation of the pump light prior to the chip in combination to edge coupling. On contrary, the lithium niobate waveguides are an elementary part of our THz generation technique. Consequently, nonlinear effects inside the TFLN chip will limit the maximal power we can use for the generation process. This aspect is investigated based on simulation in the following.

We use the python package fmas [22] that solves the unidirectional nonlinear Schrodinger equation to simulate the pulse propagation inside the

TFLN waveguides. The total dispersion is calculated starting from the effective mode refractive index that we simulated using the COMSOL Multiphysics Software. We assume a Kerr nonlinear refractive index to be the one of bulk LN: $n_2 = 1.8 \cdot 10^{-19} \text{ m}^2/\text{W}$ [6]. Furthermore, Raman scattering is expected to occur in lithium niobate. Here, we chose values of $f_R = 0.635$ for the Raman fraction, a period time of $\tau_1 = 21 \text{ fs}$ and a decay of $\tau_2 = 544 \text{ fs}$ in our simulation [23]. We note that the different values for f_R reported in the literature cause uncertainty in our simulations.

As an example, we investigate the scenario of pumping the TFLN chip (with waveguide dimensions as in our experiments) with Fourier-limited Gaussian pulses with a FWHM intensity duration of 500 fs and 1 nJ energy (10 times higher than in our experiments). Our findings reveal that using such a pulse length still contributes to optical rectification up to 500 GHz. However, shortening the pulse (increasing the peak power), would result in strong nonlinear effects. We do not find any significant change in the temporal pulse shape up to distances of 4 mm for our given waveguide geometry. Hence, we predict the terahertz generation to remain unaffected for this length (Fig. 4). In the frequency domain we can see the spectral broadening due to nonlinear effects (Fig. 4). Further, we find that pulses of 100 nJ (1000 times higher than in our experiment) propagate without significant pulse broadening over a length of 500 μm , even if strong spectral broadening occurs (Fig. 5). From these simulations, we conclude that pulse energies in the range of $\sim 100 \text{ nJ}$ will be the upper power limit. If successfully guiding a 500 fs pulse into one of our antenna gap, the amplitude of such a single-antenna emission is expected to increase by 3 orders of magnitude to 1 kV/m and the pulse energy of the THz signal can reach $\sim 10 \text{ fJ}$. ”

The content of section G is already given in the answer to comment 5 of referee 1.

3 Reviewer #3

1. The technological approach presented here is novel, the discussion is rigorous, and the experiments are well performed. The paper is nicely written and well organized. I believe that the threshold for significance required for Nature Communications is well met, and as such it meets the technical and quality standards to be accepted for publication.

I do believe, however, that the paper STRONGLY overstates the originality and capability of the technical approach relative to prior published generation methods. Specifically, the authors state that “there is a very limited leverage over tailoring the generated THz emission unless through complex pump pulse shaping, external switches or multi-pulse setups” and “the phase matching properties are fixed by the refractive index mismatch at THz and pumping frequencies in bulk crystals and can not be

controlled.” They also state “our proposed antenna schemes enable unprecedented flexibility to take control over almost all degrees of freedom of generated THz pulses.”

We thank the referee for the positive assessment of our work (its rigour, novelty and experimental data presented), including meeting the standards for publication in Nature Communications, and all of the opportunities it provides.

2. While indeed the author’s approach addresses these issues, there is substantial prior technology that in fact solved ALL these problems, enabling much higher pulse complexity than the authors present here, and with much higher generated THz power. Approaches using periodically poled crystals to achieve quasi-phase matching or arbitrary waveform synthesis were developed in the early 2000’s. Specifically, the initial demonstration in PPLN was given in

- Y.-S. Lee, et al., “Generation of Narrow-Band Terahertz Radiation via Optical Rectification in Periodically-Poled Lithium Niobate,” *Appl. Phys. Lett.* 76, 2505 (2000).

Cooling of the PPLN yielded narrow bandwidths (many more cycles in the waveform) than are presented in the present paper:

- Y.-S. Lee, et al., “Temperature Dependence of Narrow-Band Terahertz Generation from Periodically-Poled Lithium Niobate,” *Appl. Phys. Lett.* 77, 1244 (2000).

Easy tunability was demonstrated in:

- Y.-S. Lee, et al., “Tunable Narrow-Band Terahertz Generation from Periodically Poled Lithium Niobate,” *Appl. Phys. Lett.* 78, 3583 (2001).

Pulse shaping was given in:

- Y.S. Lee and T.B. Norris, “Terahertz Pulse Shaping and Optimal Waveform Generation in Poled Ferroelectric Crystals,” *J. Opt. Soc. Am B.* 19, 2791 (2002).

Highly complex waveforms (up to 53 bits, over an order of magnitude more than the 4 bits presented here), suitable for pulse coding and other sophisticated schemes, were developed in

- T. Buma and T.B. Norris, “Coded Excitation of Broadband Terahertz using Optical Rectification in Poled Lithium Niobate,” *Appl. Phys. Lett.* 87, 251105 (2005).

Finally, as it is well recognized that LN has important issues due to residual THz absorption, this approach was extended to the orientation-patterned GaAs system by the Vodopyanov and Fejer group at Stanford; see e.g.

- G. Imeshev, M. E. Fermann, K. L. Vodopyanov, M. M. Fejer, X. Yu, J. S. Harris, D. Bliss, and C. Lynch, "High-power source of THz radiation based on orientation-patterned GaAs pumped by a fiber laser," *Opt. Express* 14, 4439-4444 (2006).

The latter paper reported microwatt-level power generation. Higher power has been subsequently demonstrated.

Motivated by the comment of the referee, we now include an extensive rigorous discussion of the advantages and differences of our work compared to the previous works mentioned by the referee (and also those mentioned by referee 2.) All those works are very important for high-field generation and that in such experiments, they may be the most suited choice. While we now outline a path towards a further increase of the emitted fields by a factor of up to 20000 (making our chips interesting for mid-field applications such as spectroscopy, imaging, sensing), the novelty of our platform lies instead in its ability to deliver custom-tailored radiation from ultra-low loss on-chip waveguides. Its potential for future terahertz technologies arises from its possible integration with further chip-scale components.

We carefully rechecked the manuscript clarifying the advantages and differences of our technology in the introduction and conclusion of the main manuscript. Additionally, we added a section comparing our work do other approaches for THz generation (section G of the supplementary material, content given in answer to referee 1, comment 5) and simulated the propagation of the pump pulses inside the TFLN waveguides for different pulse energies to estimated the pump power limit (section C.2 of the supplementary material, content given in the answer to referee 2, comment 3). We hope that in its current form, our paper faithfully describes the advantages of our proposed technique and correctly discuss differences to the mentioned important prior works.

3. The emitted THz power should be directly in the abstract. THz generation technology has progressed significantly, and this critical information should be right up front in the abstract.

We follow the suggestion of the referee and now include the demonstrated achieved terahertz field amplitude in the abstract.

To address this comment from the referee, we include in the abstract: "and far-field amplitudes of few V/m"

4. Maybe it is in the text and I just missed it, but I think the crystal orientation of the LN on the chip should be clearly stated in the main text.

We provide information about the crystal cut in the main text.

To address this comment from the referee, we include in section I.A: "x-cut lithium niobate"

References

- [1] B. Desiatov, A. Shams-Ansari, M. Zhang, C. Wang, and M. Lončar. "Ultra-Low-Loss Integrated Visible Photonics Using Thin-Film Lithium Niobate." *Optica*, **6**(3):380 (2019).
- [2] W. Miao, Y. Delorme, F. Dauplay, G. Beaudin, Q. J. Yao, and S. C. Shi. "Simulation of an Integrated Log-Spiral Antenna at Terahertz." In "2008 8th International Symposium on Antennas, Propagation and EM Theory," pages 58–61. IEEE, Kunming, China (2008).
- [3] X. Wu, C. Zhou, W. R. Huang, F. Ahr, and F. X. Kärtner. "Temperature Dependent Refractive Index and Absorption Coefficient of Congruent Lithium Niobate Crystals in the Terahertz Range." *Optics express*, **23**(23):29729–29737 (2015).
- [4] Y.-S. Lee, T. Meade, M. DeCamp, T. B. Norris, and A. Galvanauskas. "Temperature Dependence of Narrow-Band Terahertz Generation from Periodically Poled Lithium Niobate." *Applied Physics Letters*, **77**(9):1244–1246 (2000).
- [5] Y. He, H. Liang, R. Luo, M. Li, and Q. Lin. "Dispersion Engineered High Quality Lithium Niobate Microring Resonators." *Optics Express*, **26**(13):16315 (2018).
- [6] Di Zhu, L. Shao, M. Yu, R. Cheng, B. Desiatov, C. J. Xin, Y. Hu, J. Holzgrafe, S. Ghosh, A. Shams-Ansari, E. Puma, N. Sinclair, C. Reimer, M. Zhang, and M. Lončar. "Integrated Photonics on Thin-Film Lithium Niobate." *Advances in Optics and Photonics*, **13**(2):242 (2021).
- [7] U. Deva and C. Saha. "Gain Enhancement of Photoconductive THz Antenna Using Conical GaAs Horn and Si Lens." In "2016 International Symposium on Antennas and Propagation (APSYM)," pages 1–3. IEEE, Cochin, India (2016).
- [8] G. Lu, R. Zhao, H. Yin, Z. Xiao, and J. Zhang. "Study of the Super Directive THz Photoconductivity Antenna." *Plasmonics*, **16**(3):677–685 (2021).
- [9] J. R. Danielson, N. Amer, and Y.-S. Lee. "Generation of Arbitrary Terahertz Wave Forms in Fanned-out Periodically Poled Lithium Niobate." *Applied Physics Letters*, **89**(21):211118 (2006).

- [10] M. Sato, T. Higuchi, N. Kanda, K. Konishi, K. Yoshioka, T. Suzuki, K. Misawa, and M. Kuwata-Gonokami. “Terahertz Polarization Pulse Shaping with Arbitrary Field Control.” *Nature Photonics*, **7**(9):724–731 (2013).
- [11] L. Gingras and D. G. Cooke. “Direct Temporal Shaping of Terahertz Light Pulses.” *Optica*, **4**(11):1416 (2017).
- [12] S. Keren-Zur, M. Tal, S. Fleischer, D. M. Mittleman, and T. Ellenbogen. “Generation of Spatiotemporally Tailored Terahertz Wavepackets by Non-linear Metasurfaces.” *Nature Communications*, **10**(1):1778 (2019).
- [13] M. Veli, D. Mengü, N. T. Yardimci, Y. Luo, J. Li, Y. Rivenson, M. Jarrahi, and A. Ozcan. “Terahertz Pulse Shaping Using Diffractive Surfaces.” *Nature Communications*, **12**(1):37 (2021).
- [14] Y.-S. Lee, T. Meade, V. Perlin, H. Winful, T. B. Norris, and A. Galvanauskas. “Generation of Narrow-Band Terahertz Radiation via Optical Rectification of Femtosecond Pulses in Periodically Poled Lithium Niobate.” *Applied Physics Letters*, **76**(18):2505–2507 (2000).
- [15] Y. S. Lee, T. Meade, T. B. Norris, and A. Galvanauskas. “Tunable Narrow-Band Terahertz Generation from Periodically Poled Lithium Niobate.” *Applied Physics Letters*, **78**(23):3583–3585 (2001).
- [16] Y.-S. Lee and T. B. Norris. “Terahertz Pulse Shaping and Optimal Waveform Generation in Poled Ferroelectric Crystals.” *Journal of the Optical Society of America B*, **19**(11):2791 (2002).
- [17] T. Buma and T. B. Norris. “Coded Excitation of Broadband Terahertz Using Optical Rectification in Poled Lithium Niobate.” *Applied Physics Letters*, **87**(25):251105 (2005).
- [18] W. L. Chan, H.-T. Chen, A. J. Taylor, I. Brener, M. J. Cich, and D. M. Mittleman. “A Spatial Light Modulator for Terahertz Beams.” *Applied Physics Letters*, **94**(21):213511 (2009).
- [19] A. Shams-Ansari, G. Huang, L. He, Z. Li, J. Holzgrafe, M. Jankowski, M. Churaev, P. Kharel, R. Cheng, D. Zhu, N. Sinclair, B. Desiatov, M. Zhang, T. J. Kippenberg, and M. Lončar. “Reduced Material Loss in Thin-Film Lithium Niobate Waveguides.” *APL Photonics*, **7**(8):081301 (2022).
- [20] G. Imeshev, M. E. Fermann, K. L. Vodopyanov, M. M. Fejer, X. Yu, J. S. Harris, D. Bliss, and C. Lynch. “High-Power Source of THz Radiation Based on Orientation-Patterned GaAs Pumped by a Fiber Laser.” *Optics Express*, **14**(10):4439 (2006).
- [21] C. Wang, C. Langrock, A. Marandi, M. Jankowski, M. Zhang, B. Desiatov, M. M. Fejer, and M. Lončar. “Ultrahigh-Efficiency Wavelength Conversion in Nanophotonic Periodically Poled Lithium Niobate Waveguides.” *Optica*, **5**(11):1438–1441 (2018).

- [22] O. Melchert and A. Demircan. “A Python Package for Ultrashort Optical Pulse Propagation in Terms of Forward Models for the Analytic Signal.” *Computer Physics Communications*, **273**:108257 (2022).
- [23] M. Bache and R. Schiek. “Review of Measurements of Kerr Nonlinearities in Lithium Niobate: The Role of the Delayed Raman Response.” (2012).

REVIEWERS' COMMENTS

Reviewer #1 (Remarks to the Author):

The revised manuscript has a better clarification on the difference and advantages of this work as compared to existing bulky LN-based THz generators. The low THz power is still the main defect, but the authors point out the ways to improve it. And It's good to see it ability to manipulate the THz polarization as well. My concerns have been carefully addressed. I recommend it for acceptance.

Reviewer #2 (Remarks to the Author):

The authors gave a sincere response to my comments and I agreed with those explanation.

I recommend publication of this paper in Nature Communications.

Reviewer #3 (Remarks to the Author):

With the revisions made in the manuscript, it is now acceptable for publication. I don't necessarily agree with all the claims made, but the paper is technically sound, and only time will tell how generally applicable this approach will be!

Response to Reviewers:

We thank all three referees for their constructive inputs in the review process and the appreciation of our work.